# Asthma and Post-Asthmatic Fibrosis: A Search for New Promising Molecular Markers of Transition from Acute Inflammation to Pulmonary Fibrosis

**DOI:** 10.3390/biomedicines10051017

**Published:** 2022-04-28

**Authors:** Innokenty A. Savin, Andrey V. Markov, Marina A. Zenkova, Aleksandra V. Sen’kova

**Affiliations:** Institute of Chemical Biology and Fundamental Medicine, Siberian Branch of the Russian Academy of Sciences, Lavrent’ev Avenue, 8, 630090 Novosibirsk, Russia; savin_ia@niboch.nsc.ru (I.A.S.); markov_av@niboch.nsc.ru (A.V.M.); marzen@niboch.nsc.ru (M.A.Z.)

**Keywords:** asthma, lung fibrosis, pulmonary fibrosis, bioinformatics analysis, microarray, in vivo model

## Abstract

Asthma is a heterogeneous pulmonary disorder, the progression and chronization of which leads to airway remodeling and fibrogenesis. To understand the molecular mechanisms of pulmonary fibrosis development, key genes forming the asthma-specific regulome and involved in lung fibrosis formation were revealed using a comprehensive bioinformatics analysis. The bioinformatics data were validated using a murine model of ovalbumin (OVA)-induced asthma and post-asthmatic fibrosis. The performed analysis revealed a range of well-known pro-fibrotic markers (*Cat*, *Ccl2*, *Ccl4*, *Ccr2*, *Col1a1*, *Cxcl12*, *Igf1*, *Muc5ac/Muc5b*, *Spp1*, *Timp1*) and a set of novel genes (*C3*, *C3ar1*, *Col4a1*, *Col4a2*, *Cyp2e1*, *Fn1*, *Thbs1*, *Tyrobp*) mediating fibrotic changes in lungs already at the stage of acute/subacute asthma-driven inflammation. The validation of genes related to non-allergic bleomycin-induced pulmonary fibrosis on asthmatic/fibrotic lungs allowed us to identify new universal genes (*Col4a1* and *Col4a2*) associated with the development of lung fibrosis regardless of its etiology. The similarities revealed in the expression profiles of nodal fibrotic genes between asthma-driven fibrosis in mice and nascent idiopathic pulmonary fibrosis in humans suggest a tight association of identified genes with the early stages of airway remodeling and can be considered as promising predictors and early markers of pulmonary fibrosis.

## 1. Introduction

Asthma is a heterogeneous pulmonary disorder with increasing worldwide prevalence, the main hallmarks of which are persistent airway inflammation, reversible airflow obstruction, and airway hyperresponsiveness [1,2]. The structural changes of the airways associated with the progression and chronization of asthma are broadly referred to as airway remodeling, characterized by cellular and extracellular changes in the large and small airways, including the increased deposition of extracellular matrix (ECM) components, such as structural proteins (collagens and elastin), adhesion proteins (fibronectin and tenascin), and glycosaminoglycans/proteoglycans [3,4,5,6,7]; the disturbance of airway epithelial cell barrier/transport/integrity, leading to goblet cell hyperplasia with mucus hypersecretion [8,9,10,11]; smooth muscle cell and fibroblast/myofibroblast proliferation [12,13,14]; as well as intensified angiogenesis in the asthmatic airways [15,16]. These signs are present in asthmatics with mild disease, but tend to worsen with advancing disease severity [3,6].

It is known that the failure of pro-resolving pathways in the early stage of asthma in patients extends the pro-inflammatory mechanisms, resulting in a chronic inflammation and irreversible changes in lung tissue associated with high mortality rates [17]. In order to successfully treat asthma and not allow its chronization, knowledge of the key markers of asthma progression is crucial. To date, clinical symptoms, such as increasing shortness of breath, chest tightness, coughing, and/or wheezing, as well as predictors of fatal asthma, such as previous episodes of near-fatal asthma, admission to an intensive care unit, and adherence to medical treatment, are usually used as prognostic signs of an unfavorable course of the disease [18]. In spite of the numerous published reports [19,20,21,22], molecular markers of asthma progression and chronization—forming a distinct regulome and not being associated with clinical symptoms, but can be revealed already in the acute phase of the disease—are still not properly identified and investigated. Given the fact that such molecular markers can be used as traits identified and targeted in the management of asthmatic patients in the early anti-fibrotic therapy, their identification and characterization are important tasks.

To date, to the best of our knowledge, a range of reports on integrative bioinformatics analysis of asthma-associated transcriptome have been published, but these works either only compare different types of acute asthma between each other (for instance, eosinophilic versus neutrophilic asthma [23] or viral versus nonviral asthma [24]) or look for marker genes related to the severity of the disorder based on a small amount of transcriptomic datasets without the proper experimental verification of the results [25,26,27].

The objectives of this study are to reveal molecular markers of the transition from acute lung inflammation to pulmonary fibrosis and to elucidate the molecular mechanisms of this process using comprehensive bioinformatics analysis of a wide set of independently obtained transcriptomics data, including functional annotation of differentially expressed genes (DEGs), reconstruction and analysis of protein–protein interaction (PPI) networks, as well as data mining analysis of revealed asthma-associated marker genes. The data obtained by in silico approaches were further validated using a murine model of ovalbumin (OVA)-induced allergic airway inflammation (hereafter OVA-induced asthma), widely used to elicit asthma-like symptoms in animals [28,29], both in the early stage of the disease and in the long-term period. The workflow of the study is shown in Figure 1.

Our findings clearly demonstrate that the structural background for the pulmonary fibrosis is already present in the subacute phase of the asthma development. Besides this, fibrosis-associated gene patterns, the expression of which significantly changes both in the acute and the late phases of asthma, were identified. Revealed key genes, including known ECM regulators, can be considered as novel promising molecular markers for fibrosis prediction in asthmatic patients.

## 2. Materials and Methods

### 2.1. GEO Datasets Analysis

The gene expression profiles associated with murine asthma were acquired from the Gene Expression Omnibus [30] database: GSE27066 (4 PBS-treated mice, 4 OVA-sensitized mice), GSE41665 (8 naïve mice, 6 OVA-sensitized mice), GSE116504 (4 naïve mice, 4 OVA-sensitized mice), GSE122197 (7 PBS-treated mice, 8 OVA-sensitized mice), and GSE50176 (4 untreated mice, 4 carbon nanotubes-treated mice). All of the analyzed datasets concern processes associated with acute asthmatic inflammation in mice at a time point earlier than 24 h after the last induction.

Additionally, the gene expression profiles of murine bleomycin-induced lung fibrosis were acquired from the Gene Expression Omnibus database: GSE8553 (3 saline-treated mice, 6 bleomycin-treated mice), GSE25640 (3 saline-treated mice, 3 bleomycin-treated mice), and GSE37635 (7 untreated mice, 6 bleomycin-treated mice). All of the analyzed datasets concern processes associated with chronic bleomycin-induced lung inflammation and pulmonary fibrosis in mice at a time point of 3 weeks after bleomycin instillation.

Gene expression profiles concerning human pathology such as idiopathic pulmonary fibrosis (GSE53845, GSE24206, GSE72073, and GSE33566), chronic obstructive pulmonary disease (GSE103174, GSE76925, GSE47460, GSE8581, GSE29133, GSE100153, GSE55962, GSE148004, GSE130928, GSE56341, GSE16972, and GSE13896), emphysema (GSE1122 and GSE26296), and cystic fibrosis (GSE38267 and GSE40445) were also acquired from the Gene Expression Omnibus database.

The descriptions of all mouse and human microarray datasets used in the study are presented in Appendix A.

The identification of differentially expressed genes (DEGs) between healthy samples and samples with pathology was performed using GEO2R, a web service that allows the comparison of two or more datasets in GEO series for DEGs identification across experimental conditions [31]. The adjusted *p*-values were applied to correct the false positive results by the default Benjamin–Hochberg false discovery rate method. Adjusted *p* < 0.05 and |fold change| > 1.5 were viewed as the cutoff values. A Venn diagram analysis of revealed DEGs was performed using the Bioinformatics and Evolutionary Genomics tool [32].).

### 2.2. Gene Set Enrichment Analysis and PPI Network Reconstruction

Functional annotation of the revealed DEGs and reconstruction of PPI networks based on the genes of interest were performed using the ClueGo v.2.5.1 plugin [33]/ToppGene web tool [34] and Search Tool for the Retrieval of Interacting Genes/Genomes (STRING) database, respectively, in Cytoscape v.3.8.1, according to the previously described procedures [35] (details in Appendix A).

### 2.3. Data Mining Analysis

To analyze the co-occurrence of genes of interest and keywords associated with lung pathology in scientific texts deposited in the MEDLINE database, a data-mining analysis of scientific literature was performed using the GenClip3 web service [36]. The list of identified DEGs common for all analyzed asthma-associated and fibrosis-associated datasets was uploaded into GenClip3 and a search of the co-occurrence of identified DEGs with the following keywords was performed: asthma, pulmonary fibrosis, lung fibrosis, hepatic fibrosis, liver fibrosis, renal fibrosis, and kidney fibrosis.

### 2.4. Mice

Female BALB/c mice with an average weight of 20–22 g were obtained from the Vivarium of Institute of Chemical Biology and Fundamental Medicine SB RAS (Novosibirsk, Russia). Mice were housed in plastic cages (9 animals per cage) under normal daylight conditions. Water and food were provided ad libitum. Experiments were carried out in accordance with the European Communities Council Directive 86/609/CEE. The experimental protocols were approved by the Committee on the Ethics of Animal Experiments at the Institute of Cytology and Genetics SB RAS (Novosibirsk, Russia) (protocol no. 56 from 10 August 2019).

### 2.5. Ovalbumin (OVA)-Induced Asthma

BALB/c mice were sensitized by intraperitoneal (i.p.) injection of a 20 μg OVA (albumin from chicken egg white, A5503, Sigma-Aldrich, St. Louis, MO, USA) and 2 mg Al(OH)_3_ (aluminum hydroxide, 239186, Sigma-Aldrich, USA) mixture suspended in 0.2 mL normal saline on days 0, 7, and 14, and then challenged with 2% OVA diluted in normal saline via aerosol inhalations on days 21, 22, 23, and 24. Inhalations were performed for 30 min by nebulizer Omron NE-C28 Plus (Omron Healthcare Co., Ltd., Kyoto, Japan) using a plexiglass induction chamber. Mice were sacrificed by cervical dislocation 24 h or 4 weeks after the last OVA challenge. The lung tissues and bronchoalveolar lavage (BAL) fluid were collected for subsequent analysis. The number of mice in each experimental group (healthy, 24 h, and 4 weeks) was nine.

### 2.6. Bronchoalveolar Lavage (BAL) Fluid Analysis

For BAL fluid procurement, the thoracic cavity of mice was opened, and the upper part of the trachea was cannulated. The lungs of mice were lavaged with 1 mL ice-cold phosphate-buffered saline (PBS). The collected BAL fluid was centrifuged at 1500 rpm for 10 min at 4 °C. The cell pellets were resuspended in 50 μL of PBS, and total leukocyte counts were performed with a Neubauer chamber after diluting in Turk solution (1:20). To determine the differential leukocyte counts (subpopulations of granulocytes, lymphocytes, and monocytes), bronchoalveolar cells were placed onto slides, stained with Azur-Eosin by Romanowsky–Giemsa staining, and examined by optical microscopy. The number of samples for BAL fluid analysis was five for each experimental group.

### 2.7. ELISA

BAL fluids were analyzed for pro-inflammatory cytokines TNF-α and IL-6 by ELISA (Thermo Scientific, Rockford, IL, USA) according to the manufacturer’s protocol (details in Appendix A).

### 2.8. Histology and Immunohistochemistry

For the histological study, the specimens of the lungs were fixed in 10% neutral-buffered formalin (BioVitrum, Moscow, Russia), dehydrated in ascending ethanol and xylol, and embedded in HISTOMIX paraffin (BioVitrum, Russia). The paraffin sections (5 μm) were sliced on a Microm HM 355S microtome (Thermo Fisher Scientific, Waltham, MA, USA) and stained with hematoxylin and eosin. The mucus production was determined using periodic acid Schiff (PAS) staining. The extracellular matrix deposition was determined using Elastica van Gieson’s and Masson’s trichrome staining. The number of samples for histological and histochemical studies was nine for each experimental group.

For the immunohistochemical study, the lung sections (3–4 μm) were deparaffinized and rehydrated. Antigen retrieval was carried out after exposure in a microwave oven at 700 W. The samples were incubated with anti-Muc5ac (ab3649, Abcam, Cambridge, MA, USA), anti-fibronectin (PA5-29578, Thermo Fisher Scientific, Waltham, MA, USA), anti-α-SMA (ab5694, Abcam, USA), anti-collagen I (MA1-26771, Thermo Fisher Scientific, USA), and anti-collagen IV (MA1-22148, Thermo Fisher Scientific, USA) primary antibodies according to the manufacturer’s protocol. Then, the sections were incubated with secondary horseradish peroxidase (HPR)-conjugated antibodies, exposed to the 3,3′-diaminobenzidine (DAB) substrate (Rabbit Specific HRP/DAB (ABC) Detection IHC Kit, ab64261, Abcam, USA), and stained with Mayer’s hematoxylin. The intensity of immunohistochemical staining of lung sections was assessed by a semi-quantitative method, where 0 is no expression, 1 is low expression, 2 is moderate expression, and 3 is high expression. The quantification was performed at a magnification of ×200 in 5–6 test fields for each lung sample; the number of samples studied was three for each experimental group.

All the images were examined and scanned using an Axiostar Plus microscope equipped with an Axiocam MRc5 digital camera (Zeiss, Oberkochen, Germany) at magnifications of ×100 and ×200.

### 2.9. Quantitative Real-Time PCR (qRT-PCR)

Total RNA was isolated from lungs of experimental animals by TRIzol Reagent (Ambion, Austin, TX, USA) according to the manufacturer’s instructions. Briefly, lung tissue was collected in 1.5 mL capped tubes, filled with 1 g of lysing matrix D (MP Biomedicals, Irvine, CA, USA) and 1 mL of TRIzol reagent, and homogenized using the FastPrep-24TM 5G homogenizer (MP Biomedicals, Irvine, CA, USA) with the QuickPrep 24 adapter. The homogenization was performed at 6.0 m/s for 40 s. After homogenization, the content of the tubes was transferred to the new 1.5 mL tubes without the lysing matrix. Total RNA extraction was carried out according to the TRIzol reagent protocol.

The first strand of cDNA was synthesized from total RNA in 100 µL of reaction mixture containing 2.5 µg total RNA, 20 µL of 5× RT buffer (Biolabmix, Novosibirsk, Russia), 250 U M-MuLV-RH revertase (Biolabmix, Novosibirsk, Russia), and 100 µM of Random Hexaprimer (5′-NNNNNN-3′). Reverse transcription was performed at 25 °C for 10 min followed by incubation at 42 °C for 60 min. Finally, reverse transcriptase was terminated at 70 °C for 10 min.

Amplification of cDNA was performed in 25 µL PCR reaction mixture containing 5 µL of cDNA; 12.5 µL of HS-qPCR (2×) master mix (Biolabmix, Novosibirsk, Russia); 0.25 µM each of forward and reverse primers to *HPRT-* and *HPRT*-specific ROX-labeled probes; and 0.25 µM each of forward and reverse gene-specific primers and FAM-labeled probes (Appendix A). Amplification was performed as follows: (1) 94 °C, 2 min; (2) 94 °C, 10 s; (3) 60 °C, 30 s (50 cycles). The relative level of gene expression was normalized to the level of *HPRT* according to the ∆∆Ct method.

The relative level of gene expression was determined with the CFX96^TM^ Real-Time system (C1000 Touch^TM^, Hercules, CA, USA).

Three to five samples from each experimental group were analyzed in triplicate.

The sequences of primers used in the study are listed in Appendix A.

### 2.10. Statistical Analysis

Statistical analysis was performed using the following methods. The correction of the false positive results by the default Benjamin–Hochberg false discovery rate method during the identification of differentially expressed genes was carried out through the GEO2R web-tool. Significance in the functional enrichment analysis was determined using a two-sided hypergeometric test with Bonferroni step-down corrections in the ClueGo module of Cytoscape software. In all other cases (leukocytes counting, cytokines evaluation, IHC scoring, and qRT-PCR assay), statistical analysis was performed using two-tailed unpaired Student’s *t*-test in Microsoft Excel; *p*-values of less than 0.05 were considered statistically significant. The data are expressed as the mean ± SD.

## 3. Results

### 3.1. Identification of Key Genes Involved in the Development of Ovalbumin (OVA)-Induced Asthma and Post-Asthmatic Fibrosis

#### 3.1.1. Key Genes Associated with Acute Asthma

In order to identify core genes associated with asthmatic inflammation and, potentially, post-asthmatic fibrosis in lungs, the changes in the gene expression patterns between asthmatic and healthy lungs in OVA-treated and untreated mice were assessed by reanalysis of cDNA microarray data from GSE41665, GSE116504, GSE122197, and GSE27066 datasets deposited in the Gene Expression Omnibus (GEO) database, using the GEO2R tool (Figure 2). Moreover, given the fact that inhalation of rigid multiwalled carbon nanotubes in mice induces innate immunity-mediated allergic-like inflammation in lungs [37], the GSE50176 transcriptomic dataset from this study was also included in our analysis. In order to unify the selected GEO datasets, only transcriptomic profiles of murine lungs collected 24 h after the last OVA or nanotubes challenge were analyzed.

Venn diagram analysis of differentially expressed genes (DEGs) (|fold change| > 1.5, *p* < 0.05) revealed 47 DEGs common for all analyzed datasets (Figure 2A). Functional annotation of this gene set showed significant enrichment of terms associated mostly with the regulation of the chemokine-mediated signaling pathway, eosinophil chemotaxis, and lung epithelial cell differentiation, the well-known processes involved in the regulation of asthma [39], as well as the biosynthesis of specialized pro-resolving lipid mediators, playing an important role in the clearance and regulation of inflammatory exudates [40] (Figure 2B). In order to better understand the relation between 47 common DEGs, a protein–protein interaction (PPI) network was constructed using the Search Tool for the Retrieval of Interacting Genes/Proteins (STRING) database, retrieving interactions of nodes within the network with a sufficiently high confidence level (confidence score ≥ 0.7) (Figure 2C). It was found that only 10 genes out of 47 revealed asthma-specific DEGs formed a loosely coupled network (Figure 2C), whereas the remaining DEGs did not have interconnections with each other at all. The obtained results demonstrate that identified asthma-related genes can be involved in different compartments of the asthma-associated regulome.

Next, in order to more thoroughly characterize the revealed asthma-specific DEGs, their expression profiles were clustered using Euclidean distances, and two main clades, containing 34 upregulated and 13 downregulated DEGs, were identified (Figure 2D). Further analysis of their degree centrality scores in PPI networks reconstructed for each analyzed GEO datasets demonstrated a significant interconnection of revealed key DEGs with the asthma-associated regulome that can indicate their probable modulating role in the pathogenesis of asthma (Figure 2E). The most central (top-10) hub genes in the analyzed networks included:Genes encoding pro-inflammatory chemokines (*Ccl6*, *Ccl9*, *Ccl12*), responsible for the chemotaxis of monocytes and T cells, taking part in a wide spectrum of inflammatory diseases, including allergic airway inflammation [41,42,43,44];An inhibitor of matrix metalloproteinase *Timp1*, which exacerbates a range of pulmonary disorders associated with inflammation, fibrotic, and malignant transformation [45,46,47,48,49];Components of airways mucus *Muc5ac*, an imbalance of which takes part in the development of asthmatic inflammation, chronic obstructive pulmonary disease (COPD), and lung fibrosis [50,51,52,53,54];Coagulation factor *F5*, participating in the activation of prothrombin to thrombin and, thus, implicated in the development of asthmatic lung inflammation [55];Cytochrome *Cyp2e1*, a member of cytochrome P450 family, involved in the xenobiotics metabolism and asthma development [56];Adrenergic receptor *Adra2a* and transcriptional factor *Ear1*, involved in the regulation of the sympathetic nervous system [57] and metabolic processes [58], respectively; andThymidine kinase 1 *Tk1*, a ubiquitous enzyme required for DNA damage repair and used as a biomarker for diagnostic and prognostic stratification of various cancers, including lung adenocarcinoma [59,60].

Interestingly, 6 of the 10 revealed asthma-specific hub genes (*Ccl6*, *Ccl9*, *Ccl12*, *Timp1*, *Muc5ac*, *Cyp2e1*) were involved in rodent inflammatome, a gene signature identified previously by Wang et al. in 11 independent rodent inflammatory disease models [61], clearly showing their key regulatory function in acute inflammation (Figure 2E).

Additionally, we questioned how substantially the identified 47 asthma-related DEGs have been studied in the field of asthma and its chronization. To understand this, an analysis of co-occurrence of their human ortholog gene names and “asthma” or “pulmonary fibrosis” keywords within the scientific texts deposited in the MEDLINE database was performed using the GenCLiP3 text mining tool. The obtained results demonstrated that only 26 of 47 studied genes have been associated with mentioned respiratory disorders (Figure 2F), involving 9 of the top-10 hub genes revealed above by the network analysis (the most associated: *Muc5ac*, *Ccl12* (human ortholog *CCL2*), *Muc5b*, *Timp1*; studied to a lesser extent: *Ccl6* and *Ccl9* (human ortholog *CCL15*), *Cyp2e1*, *F5*, and *Adra2a*). Our findings, firstly, show the adequate reliability of the results obtained by an in silico approach (revealed hub genes have already been widely studied in the field of asthma pathogenesis (Figure 2F)); secondly, they can be considered as a source of novel molecular markers of asthma and related disorders; and, thirdly, they clearly demonstrate that several acute asthma-associated key nodes, such as *Muc5ac*, *Ccl12*, *Timp1*, etc., can also be connected with the development of pulmonary fibrosis (Figure 2F).

#### 3.1.2. Key Asthma-Specific Genes Associated with Post-Asthmatic Fibrosis in Lungs

In the next step of the study, we probed whether obtained asthma-related DEGs are associated with pulmonary fibrosis. To answer this question, the PPI network consisting of 10 revealed key genes with their first neighbors extracted from analyzed asthma-related regulomes (common for at least three of five used GEO datasets) was created (Figure 3A). It was found that the reconstructed network contained four distinct modules: the first one involved 7 of the 10 investigated key genes (*Timp1*, *F5*, *Ccl12*, *Ccl6*, *Ccl9*, *Adra2a*, and *Ear1*) with their gene partners, whereas the other three modules were centered on unique hub genes, namely, *Tk1*, *Muc5ac*, and *Cyp2e1* (Figure 3A). Such architecture of the network indicates that the revealed key nodes are related to four different compartments of the asthma-associated regulome and some of them (*Timp1*, *F5*, *Ccl12*, *Ccl6*, *Ccl9*, *Adra2a*, and *Ear1*) can be involved in the regulation of similar processes.

Further gene set enrichment analysis of the top-10 key nodes and their partner genes demonstrated their association with functional terms related not only to acute asthma (eosinophil chemotaxis, neutrophil degranulation, chemokine activity and chemokine-mediated signaling pathway, blood coagulation, acute inflammatory response, regulation of cytokine production, MAPK cascade, and mucins glycosylation), but also to lung remodeling and fibrosis-related processes (notably, lung fibrosis itself, as well as processes associated with cell proliferation, chemotaxis, migration, and adhesion; extravasation and angiogenesis; endothelial/epithelial cell proliferation; smooth muscle cell proliferation/migration; organ and tissue regeneration; regulation of wound healing; ECM enzymes and growth factors activity) (Figure 3B, Appendix A).

In order to examine this relation more rigorously, and to identify probable markers of the transition of acute asthmatic inflammation to post-asthmatic fibrosis among analyzed genes (Figure 3A), 103 selected asthma-related DEGs (top-10 key nodes and their 93 first neighbors) were overlapped with key genes involved in pathogenesis of pulmonary fibrosis extracted from DisGeNET database (Figure 1, left branch of the diagram). Venn diagram analysis revealed 24 genes common for both disorders, including asthma-specific key nodes *Timp1*, *Cyp2e1*, and *Muc5ac* (Figure 3C). Further exploration of asthma-associated PPI networks demonstrated that all 24 selected genes were characterized by a high degree of centrality and, therefore, can be involved in asthma pathogenesis (Figure 3D). The list of these genes included DEGs, encoding (a) pro-inflammatory cyto/chemokines as well as their receptors (*Ccr3*, *Ccl4*, *Ccr2*, *Cxcl12*, *Ccl2*); (b) regulators of extracellular matrix (*Fn1*, *Timp1*, *Thbs1*, *Tnc*, *Mmp2*, *Spp1*); (c) components of complement (*C3*, *C3ar1*, *Cfd*,); (d) insulin-like growth factor and its receptor (*Igf1*, *Igfbp3*); (e) mucins, the main components of pulmonary mucus (*Muc5ac*, *Muc5b)*; (f) antioxidant proteins (*Cat* and *Cyp2e1*); and (g) regulators of bronchoconstriction (*Cysltr1*), innate immunity (*Arg1*), blood coagulation (*F10*), and smooth muscle constriction (*Grp*).

In order to clarify whether the mentioned key genes have been previously studied as fibrosis-related ones, a text mining approach was used. Given well-known convergence in the fibrotic remodeling processes in different organs [62], we searched published data for the associations of listed genes not only with pulmonary fibrosis, but also with fibrotic changes in liver and kidney (Figure 3E). As shown in Circos plots depicted in Figure 3E, almost all the selected genes were related to mentioned disorders to some extent. It was found that *Ccl2*, *Ccr2*, *Cxcl12*, *Timp1*, *Igf1*, *Spp1*, and *Cat* were interconnected with fibrosis of all three organs and, therefore, can be considered as probable pan-fibrotic markers; *Ccl4*/*Thbs* were associated with fibrotic changes in both liver and kidney; and *Muc5b*/*Muc5ac/Grp*, *Cyp2e1*, and *C3* were more specific for pulmonary, hepatic, and renal fibrosis, respectively (Figure 3E).

Finally, in order to assess how deeply identified genes are interconnected with pulmonary fibrosis regulome, a PPI network was retrieved from all analyzed lung fibrosis-associated genes collected from the DisGeNET database, followed by the computing of the degree centrality scores of each evaluated 24 asthma/fibrosis-specific DEGs within the reconstructed network (Figure 3F). The obtained results show that the majority of analyzed genes were highly interconnected with fibrotic regulome, among which *Fn1*, *Igf1*, *Ccl2*, *C3*, and *Timp1* were identified as the most influential nodes (Figure 3F).

Given the high association of genes listed in Figure 3F with both asthma and pulmonary fibrosis, the top-10 asthma-related genes the most interconnected with pulmonary fibrotic regulome (*Fn1*, *Igf1*, *Ccl2*, *C3*, *Timp1*, *Cxcl12*, *Ccl4*, *Ccr2*, *Spp1*, *C3ar1*); *Cat*, playing an important role in fibrosis of various organs [63,64,65]; as well as *Cyp2e1* and *Muc5ac*/*Muc5b*, more specific for liver and lung fibrosis, respectively, were selected for further validation in a murine model of OVA-induced asthma.

### 3.2. Ovalbumin (OVA)-Induced Asthma Model

Given the fact that the key points of airway remodeling and fibrogenesis in asthmatic lungs cannot be fully explored in patients, murine models of asthma are crucial for understanding the disease mechanisms and identification of promising molecular markers and potential therapeutic targets of this pathology. Among them, acute asthma driven in mice by OVA/Al(OH)_3_ sensitization with a subsequent single cycle of OVA challenge is a classical model reflecting the initial events of the disease, mainly acute airway inflammation [29]. Further increase in the number of OVA challenge cycles in mice can contribute to the development of irreversible long-term effects in asthmatic airways, such as airway remodeling and fibrosis [66,67].

We studied the development of OVA-induced asthma both in the acute phase and remote period (i.e., subacute phase) before the progression of irreversible changes in the lungs [21,68,69]. BALB/c mice were sensitized with three-fold subsequent intraperitoneal (i.p.) injections of a 20 μg OVA/2 mg Al(OH)_3_ mixture at days 1, 7, and 14, and then challenged with 2% OVA via four consecutive aerosol inhalations at days 21, 22, 23, and 24 (Figure 4A). Samples for BAL fluid, histological, and qRT-PCR analysis were collected 24 h (day 25, acute phase) and 4 weeks (day 52, subacute phase) after the last OVA challenge (Figure 4A).

It is well known that asthma progression is accompanied by airway remodeling traits such as inflammatory infiltration, mucus overproduction, and ECM protein deposition [66,70]. As shown in Figure 4B, aerosol inhalations of OVA caused the development of acute inflammation in the respiratory system of mice, characterized by a nearly 44-fold increase in the number of total leukocytes in the BAL fluid, represented predominately by granulocyte subpopulation and accompanied by 2.3- and 3.9-fold increases in the TNF-α and IL-6 levels, respectively, compared to healthy animals (Figure 4C). Histologically, the lung tissue of OVA mice 24 h after the last challenge is mainly characterized by the inflammatory infiltration, represented by granulocytes with a monocyte/macrophage admixture and located around the bronchi (Figure 4D). Mucus overproduction, which is a pathognomonic sign of asthma [50,71] identified by the PAS staining, was detected in the bronchial epithelium of OVA mice in the acute phase of lung inflammation (Figure 4D). Special staining for ECM deposition, such as Elastica van Gieson and Masson trichrome staining, showed basal staining of the lung stroma both in healthy animals and mice with asthma 24 h after the last OVA challenge (Figure 4D). Immunohistochemical staining with anti-α-SMA primary antibodies also showed basal staining of stromal structures of lungs and revealed no significant differences between healthy mice and OVA mice 24 h after the last challenge (Figure 4D).

The subacute phase of OVA-induced lung inflammation evaluated 4 weeks after the last challenge is characterized by the decrease in the number of leukocytes and granulocytes as well as IL-6 in the BAL fluid, but not to healthy levels, while TNF-α reached the values of healthy animals (Figure 4B,C). Histologically, lung tissue showed a residual inflammatory infiltration, as well as decrement in the mucus secretion in the bronchial epithelium (Figure 4D). However, evaluating long-term signs of lung tissue remodeling, extensive collagen deposition and incipient peribronchial and pulmonary fibrosis were found in the lungs of OVA mice already 4 weeks after the last challenge in the absence of repeated OVA induction cycles, which, according to the literature, are necessary for the development of chronic inflammation [66]. As shown by Masson’s trichrome and Elastica van Gieson staining, collagen and elastic fiber content increased significantly in the lung parenchyma and airways of mice in the subacute phase compared to the healthy mice and mice in the acute phase of OVA-induced lung inflammation (Figure 4D). Immunohistochemistry (IHC) of the lung sections indicated a strong increase in the expression of α-SMA in the lungs of mice 4 weeks after the last OVA challenge compared with healthy mice and mice 24 h after the last challenge, which indicates a reliable pulmonary fibrosis (Figure 4D).

According to published data, the development of fibrotic changes in lung tissues of mice with OVA-driven asthma requires several allergen challenge cycles in sensitized animals during long time periods [66,72]. In this study, we showed for the first time reliable initiation of fibrotic changes in the lungs of OVA-treated mice in the absence of recurring OVA cycles. Thus, the setting of the OVA-induced asthma murine model at the used time points demonstrates a clinical and histological picture of the early stages of airway remodeling and lung fibrosis formed already 4 weeks after the last OVA induction, which allows this model to be used to study the molecular mechanisms associated with both acute asthma and post-asthmatic pulmonary fibrosis.

### 3.3. Analysis of Gene Expression Patterns Associated with Asthma and Pulmonary Fibrosis Development

The expression levels of key genes associated with both acute asthma and pulmonary fibrosis development (*Fn1*, *Igf1*, *Ccl2*, *C3*, *Timp1*, *Cxcl12*, *Ccl4*, *Ccr2*, *Spp1*, *C3ar1*, *Cat*, *Cyp2e1*, *Muc5ac*, and *Muc5b*), revealed by bioinformatics analysis, were evaluated in the lung tissue of OVA-challenged mice 24 h (asthmatic lungs) and 4 weeks (fibrotic lungs) after last OVA exposure by TaqMan-based qRT-PCR. In addition, the expression of markers reflecting both the persistence of inflammation (Muc5ac) and fibrosis development (Fn1) was validated at the protein level using IHC. The obtained results are summarized in Figure 5 and Table 1.

Lung tissue of healthy animals was characterized by low expression levels of studied genes, which were set as 1, whereas OVA challenge caused a manifold increase in their expression (Table 1, Figure 5A). It was found that the magnitude of expression changes in asthmatic lungs (24 h after the last OVA administration) was much more pronounced compared to the samples of fibrotic lungs (4 weeks after the last OVA administration) (Figure 5A). The fold changes of the gene expression between healthy control and OVA-challenged mice in these two time points were as follows (in descending order): *Timp1* (~45-fold in asthmatic lungs and 2.3-fold in fibrotic lungs) > *Ccl2* (~29- and 2.3-fold) > *Igf1* (~16- and 1.7-fold) > *Muc5b* (~12- and 2-fold) > *C3* (~10- and 3.3-fold) > *Fn1* (5.9- and 2.3-fold). *Spp1*, *C3ar1*, and *Muc5ac* were found to be significantly upregulated only in asthmatic lungs (4.7-, 3.4-, and 1.8-fold, respectively), whereas these genes were expressed at the same level in fibrotic and healthy lungs (Figure 5A). The expression of *Cxcl12*, *Ccl4*, and *Ccr2* was insignificantly increased in asthmatic lungs and reached levels similar to healthy mice in fibrotic lung tissues (Figure 5A). Finally, *Cyp2e1* and *Cat*, DEGs downregulated in analyzed asthma-related GEO datasets (Figure 2), were significantly suppressed both in asthmatic and fibrotic lungs: *Cat* by 11.2- and 9.6-fold, respectively, and *Cyp2e1* by 4.8- and 2.8-fold, respectively, compared with healthy control (Figure 5A). As can be seen from Table 1, the fold change in the expression of genes validated in vivo tends to be similar to the same ones in microarray datasets.

In the next step, the protein levels of Fn1 and Muc5ac in healthy, asthmatic, and asthma-driven fibrotic lungs were evaluated by IHC staining. As shown in Figure 5B, no significant differences in Fn1 expression were found between healthy and asthmatic samples (the basal staining of bronchial epithelium surface and lung stroma for Fn1 was detected in both mentioned groups), whereas lung tissues sampled 4 weeks after the last OVA exposure were characterized by a slight increase in the expression of Fn1 in comparison with both healthy and asthmatic mice (Figure 5B,C). Protein expression of Muc5ac was significantly upregulated both in asthmatic and fibrotic lungs compared to healthy controls, especially in the interalveolar septum area (Figure 5B,C). The obtained results are generally consistent with data of qRT-PCR analysis: *Fn1* mRNA level was found to be slightly increased in fibrotic lung tissue and expression of *Muc5ac* mRNA was significantly upregulated in asthmatic lungs in comparison with healthy counterpart (Figure 5A). Interestingly, IHC analysis did not confirm the significant upregulation of Fn1 in asthmatic lungs and the drop in Muc5ac expression in fibrotic lungs revealed by qRT-PCR (Table 1). The observed discrepancy between mRNA and protein levels of Fn1 is not surprising: previously, a similar ratio of Fn1 (high mRNA and low protein levels) was observed in the acute phase of CCl_4_-induced liver injury in rats preceding liver fibrosis [73]. As for Muc5ac, the revealed imbalance between mRNA and protein expression can be associated both with a slowdown of protease-dependent mucus degradation and an impairment of mucociliary clearance in acute asthmatic airway inflammation [74,75]. Given the fact that a wide range of parameters can influence mRNA–protein correlation, including the effects of translational modulators, ribosome occupancy or protein turnover [76], the verification of revealed gene markers of asthma-driven fibrosis at protein level is an intricate topic for further investigation.

Thus, performed qRT-PCR analysis and IHC studies, on the one hand, clearly demonstrate the credibility of performed bioinformatics analysis, and, on the other hand, show that 9 of 14 evaluated asthma-specific key nodes are significantly up- (*Fn1*, *Igf1*, *Ccl2*, *C3*, *Timp1*, *Muc5b*, *Muc5ac*) or downregulated (*Cat*, *Cyp2e1*) not only in asthmatic, but also in fibrotic lung tissues, indicating their probable associations with early fibrotic changes in the respiratory tract already at the stage of asthma-driven inflammation.

### 3.4. Overlapping of Key Genes Related to Acute Asthma and Asthma-Driven Fibrosis with Genes Associated with Bleomycin-Induced Lung Fibrosis

In the next step of the study, we questioned whether revealed key genes specific for asthma and post-asthmatic fibrosis play a regulatory role in the development of pulmonary fibrosis of a non-allergic etiology (Figure 1, right branch of the diagram). To understand this, a bleomycin-induced lung fibrosis model was chosen, since bleomycin is known to cause fibrotic changes due to cytotoxicity and lung injury [77]. Reanalysis of three independent transcriptomic datasets related to bleomycin-driven fibrosis revealed 130 common bleomycin-specific DEGs playing probable regulatory functions in this disorder (Figure 6A, upper panel). Further overlapping of these genes with the core asthma-related gene network (Figure 6A, bottom panel) demonstrated that 11 asthma-annotated genes, including three key nodes (*Ccl6*, *Ccl9*, *Timp1*) (Figure 2E) and five genes displaying high interconnections with DisGeNET-retrieved lung fibrosis regulome (*Igf1*, *Timp1*, *Spp1*, *Tnc*, *Mmp12*) (Figure 3F), were associated with the development of bleomycin-induced fibrotic changes in lungs (Figure 6A), which indicates a possible universality of the pathways regulating pulmonary fibrosis of different etiologies.

Next, in order to evaluate the significance of revealed bleomycin-specific DEGs (119 in Figure 6A, bottom panel) as universal fibrotic markers, the nodal genes (23 out of 119) were selected from this gene set according to the following criteria: (a) degree > 20, (b) consistent expression in all bleomycin-related datasets (either up- or downregulation, but not a mix of them), and analyzed by text-mining (Figure 6B). The obtained results show that regulators and components of extracellular matrix (*Mmp2*, *Mmp9*, *Col1a1*, *Eln*, *Col3a1*) as well as protein phosphatase *Pten* were quite thoroughly evaluated in fibrosis field and can be considered as promising pan-fibrotic markers (Figure 6B), whereas for other nodal genes (*Apoe*, *Fcgr3*, *Top2a*, *Ctss*, *Itgb6*, *Thbs2*, *Col4a1*, *Csrp3*, *Tyrobp*, *Fcgr2b*, *Lamc2*, *Ttn*, *C1qa*, *Col4a2*, *Actn2*, *Fcer1g*, *Atpa2a*), low association with this disorder was found in published scientific articles (Figure 6B), which shows the expediency of their further investigation as novel fibrotic marker genes.

To estimate the expression level of bleomycin-specific key genes in the lungs of mice with asthma and asthma-associated fibrosis, pan-fibrotic regulator *Col1a1* as well as *Col4a1*, *Col4a2*, *Thbs2*, and *Tyrobp*—novel pulmonary fibrosis marker candidates poorly investigated as lung fibrosis-related genes and associated only with hepatic or renal fibrosis—were chosen for qRT-PCR analysis. It was found that expression of *Col1a1*, *Col4a1*, *Col4a2*, *Thbs2*, and *Tyrobp* was upregulated by 16.8-, 6.7-, 3.9-, 2.9-, and 1.5-fold in asthmatic lungs compared with healthy controls, respectively (Figure 6C). The chronization of asthma and development of lung fibrosis was shown to either not affect observed asthma-driven upregulation of the genes, as in the case of *Col1a1*, *Col4a1*, and *Col4a2*, or downregulate the expression of *Thbs2* and *Tyrobp* almost to the level of healthy control (Figure 6C).

Since *Col4a1* and *Col4a2* not yet related to pulmonary fibrosis according to the text mining analysis (Figure 6B) were highly interconnected with bleomycin-induced fibrosis regulome and upregulated in asthmatic and asthma-driven fibrotic lungs (Figure 6C), their expression along with the expression of *Col1a1* were further verified on the protein level using IHC staining. As shown in Figure 6D,E, stromal structures of healthy and asthmatic lungs were stained with anti-collagen I and anti-collagen IV antibodies at similar basal levels, whereas fibrotic lungs were characterized by significant upregulation of collagens I and IV expression compared to healthy control. The obtained data agreed well with the results described above, which demonstrated the development of reliable pulmonary fibrosis in OVA-treated mice 4 weeks after the last challenge (Figure 4D). The revealed discrepancy between expressions of collagens I and IV at mRNA and protein levels in mice with asthma is in line with the mRNA/protein ratio of Fn1 observed in asthmatic lungs (Figure 5). Apparently, upregulation of type IV collagen in lungs is more specific for the initial stages of fibrotic process: recent proteome analysis demonstrated, in agreement with our findings, strong upregulation of COL4A1 in the latest stage of asthma in mice [78], and Urushiyama et al. showed massive production of type IV collagen by α-SMA-positive myofibroblasts during early fibrotic lesions of idiopathic interstitial pneumonias in humans [79].

Given the contradictory role of type IV collagen in pulmonary fibrosis development [80,81,82], we next questioned how deeply *Col4a1* and *Col4a2* are associated with a regulation of fibrosis-related processes. Comparison of interactome of *Col4a1* and *Col4a*2 with interactome of *Col1a1*, a key molecular determinant of pulmonary fibrosis [83], revealed their significant similarity—approximately 77% of partner genes of *Col4a1* and *Col4a2* retrieved from STRING database were found to be first neighbors of *Col1a1* (Figure 7A). Further functional annotation of neighboring genes being unique for *Col4a1/2* also demonstrated their tight association with fibrosis-related processes, including synthesis, organization, and degradation of ECM; biosynthesis of collagens; cellular adhesion and motility; fibroblast proliferation; and regulation of wound healing (Figure 7B,C). Moreover, our analysis of the DisGeNET database demonstrated that *Col4a1/2* and their unique partner genes are involved in the regulation of a range of disorders accompanied by fibrotic lesions, such as Alport syndrome, renal glomerular disease, fibrosing alveolitis, cirrhosis, as well as pulmonary fibrosis itself (Figure 7C).

Thus, the performed analysis clearly demonstrates possible involvement of revealed asthma-related key genes in the regulation of lung fibrosis of different etiologies and evidence the expediency of further investigation of type IV collagen as a novel promising marker of early lung fibrosis.

## 4. Discussion

Chronic respiratory diseases are the third leading cause of death, behind cardiovascular diseases and cancer, affecting approximately 550 million people all over the world [84]. Most of the chronic respiratory diseases are attributable to asthma and chronic obstructive pulmonary disease (COPD), the latter being the major cause of death [85,86]. Despite differences in etiology and symptoms, a common feature of asthma and COPD is airway inflammation resulting in pulmonary fibrosis [87]. The severity of asthma- and COPD-related inflammation might differ in diverse respiratory conditions and effectiveness of pharmacological anti-inflammatory treatments and, thus, predicting fibrosis development is unlikely to be equal in all patients [84,88,89]. A precision medicine approach based on an understanding of the molecular mechanisms involved in asthma progression and fibrosis is required to increase the chance of therapeutic success in patients.

An extensive bioinformatics analysis with subsequent validation of the revealed key genes on the in vivo model of OVA-induced asthma was performed to identify marker genes and pathways associated with asthma progression and lung fibrosis development. In the first step, we analyzed cDNA microarray datasets connected with acute allergic airway inflammation and revealed top-10 core genes (*Ccl6*, *Ccl9*, *Ccl12*, *Timp1*, *Muc5ac*, *F5*, *Cyp2e1*, *Adra2a*, *Ear1*, *Tk1*) characterized by (a) nodal disposition within asthma-related gene networks, (b) association with the well-known processes involved in the regulation of asthma, and (c) close connection both with “asthma” and “pulmonary fibrosis” terms in the scientific texts (Figure 2). Overlapping the identified top-10 key nodes and their first neighbors with DisGeNET pulmonary fibrosis-associated genes allowed us to reveal possible markers of the transition of acute asthmatic inflammation to post-asthmatic fibrosis (Figure 3). The most influential nodes (*Fn1*, *Igf1*, *Ccl2*, *C3*, *Timp1*, *Cxcl12*, *Ccl4*, *Ccr2*, *Spp1*, *C3ar1*, *Cat*, *Cyp2e1*, *Muc5ac*, *Muc5b*) were selected for further validation in the OVA-induced asthma and post-asthmatic fibrosis murine model (Figure 3F).

Validation of revealed key asthma/fibrosis-associated genes in our murine model of OVA-driven asthma and asthma-associated fibrosis clearly confirmed the reliability of obtained bioinformatics data: 11 out of 14 analyzed genes were differentially expressed in asthmatic lungs (upregulated: *C3*, *C3ar1*, *Ccl2*, *Fn1*, *Igf1*, *Muc5ac*, *Muc5b*, *Spp1*, *Timp1*; downregulated: *Cat*, *Cyp2e1*), and eight fibrosis-related nodal genes were found to significantly change their expression in fibrotic lungs (upregulated: *C3*, *Ccl2*, *Fn1*, *Igf1*, *Muc5b*, *Timp1*; downregulated: *Cat*, *Cyp2e1*) compared to healthy counterparts (Figure 5A). Interestingly, the expression of revealed key upregulated DEGs reached a maximum in the acute phase of inflammation, whereas pulmonary fibrosis development was accompanied by decrease in their expression either to significantly lesser values (*C3*, *Ccl2*, *Fn1*, *Igf1*, *Muc5b*, *Timp1*) or to the level of healthy controls (*C3ar1*, *Muc5ac*, *Spp1*) (Figure 5A). Despite observed downregulation during fibrotic remodeling, the nodal positions of evaluated genes within the fibrosis-associated PPI network (Figure 3F) demonstrated their probable involvement in the regulation of early fibrotic changes in lungs already at the stage of acute/subacute inflammation. In the case of *Cyp2e1* and *Cat* suppressed in asthmatic lungs and highly interconnected with asthmatic (Figure 2E) and fibrotic (Figure 3F) regulomes, respectively, their expression in fibrotic lungs also remained significantly reduced compared to healthy control (Figure 5A).

The comparison of gene markers of asthma and asthma-associated pulmonary fibrosis with key genes of allergy-independent bleomycin-induced fibrosis demonstrated low interconnection of asthma-related DEGs with bleomycin-perturbed regulome in murine lungs—only 11 genes were identified as common for both asthma/post-asthmatic fibrosis and bleomycin-induced fibrosis (Figure 6A, lower Venn diagram), among which three genes (*Igf1*, *Spp1*, *Timp1*) displayed the most obvious association with fibrotic process in lungs. The obtained results agree well with published data—indeed, the proteins encoded by these genes are known regulators of fibrosis. For instance, insulin-like growth factor 1 (IGF1) was found to induce core fucosylation of proteins in alveolar epithelial cells with subsequent senescence of the latter [90] and be required for myofibroblast differentiation [91]. Osteopontin encoded by *Spp1* can activate matrix metalloproteinase 7 (MMP-7) and enhanced motility and proliferation of fibroblasts [92] as well as is involved in the regulation of epithelial mesenchymal transition (EMT) in lungs of bleomycin-treated mice [93]. Metalloproteinase inhibitor TIMP1 was found to be involved in the maintenance of a balance between proteases and anti-proteases in lungs, the impairment of which leads to idiopathic pulmonary fibrosis development [94], and regulates the proliferation of fibroblasts by direct interaction with CD63 and integrin β1 exposed on their surface [95].

The performed complex analysis of murine regulome of OVA-induced asthma revealed key genes associated not only with acute phase of the disorder, but also with fibrotic remodeling in asthmatic lungs, the summarized network of which is depicted in Figure 8. As shown in the network, a number of revealed asthma-specific genes plays known regulatory functions in pulmonary fibrosis, such as *Igf1*, *Spp1*, and *Timp1* mentioned above; mucins *Muc5ac* and *Muc5b*, playing a key role in resolving fibrosis-induced disrupted homeostasis in lungs [96]; chemokines *Ccl2* and *Ccl12* involved in recruiting fibrocytes and exudative macrophages in fibrotic areas [97]; as well as catalase (*Cat*), inhibiting fibrosis-associated mitochondrial DNA damage in alveolar epithelial cells and their apoptosis [98]. Besides this, our analysis identified several genes, on the one hand, highly interconnected with fibrotic regulome, and, on the other hand, poorly investigated as probable regulators of pulmonary fibrosis (Figure 8). Given these findings, the following genes and their protein products can be considered as novel potential markers of asthma-associated lung fibrosis:Fibronectin (Fn1), which mediates cell–matrix adhesion by binding ECM proteins [99]; immobilizes latent TGF-β-binding protein-1 (LTBP-1) involved in the activation of TGF-β [100]; stimulates EMT of alveolar epithelial cells [101]; mediates fibroblast invasion being exposed on the surface of extracellular vesicles [102]; and correlates with asthma severity via coagulation impairment and prothrombotic properties [103].Complement C3 and its receptor C3ar1, involved in the response to multiple types of lung injury [104,105,106], the increased expression of which was revealed in lungs of mice and humans with bleomycin-induced fibrosis [107] and pulmonary fibrosis [108], respectively. Activated complement C3a was shown to stimulate human epithelial lung cells to express TGF-β1 [107] and pharmacological blockade of C3ar1 attenuated the development of pulmonary fibrosis in bleomycin-treated mice [108].Thrombospondin-1 (Thbs1), which binds to and activates latent TGF-β [109] and participates in collagen trafficking, processing, and fibril assembly [110].Chemokine receptor Ccr3, playing a key role in the accumulation and activation of eosinophils in the allergic airways [111], the neutralization or knockout of which caused significant reduction in the severity of bleomycin-induced fibrosis in mice [112] and fibrosis-associated airway remodeling in IL13 transgene-induced mice model [113]. Moreover, it was found that high expression levels of CCR3 are associated with worse survival outcomes in patients with pulmonary fibrosis [114].Cytochrome Cyp2e1, the expression of which, in agreement with our data, was generally decreased during chronic inflammation in mice, including bronchial asthma [115] and liver fibrosis induced by CCl4 and bile duct ligation [116], and patients with fibrotic lung diseases [117].Chemokines Ccl6 and Ccl9 (human ortholog CCL15), showing a chemotactic effect on blood monocytes and eosinophils in severe asthma [118] and mediating the motility of fibrocytes [119]. Elevated levels of CCL15 and Ccl9 were found in lungs of patients and mice with chronic hypersensitivity pneumonitis [120] and chronic graft-versus-host disease [121], respectively.Tenascin-C (Tnc), the direct binding of which with integrin αvβ1 on the surface of human fibroblasts led to the activation of the TGF-β/Smad2/3 signaling axis followed by differentiation of the cells to myofibroblasts [122] and activation of type I collagen production [123].Integrin αvβ6 (Itgb6), involved in the activation of latent TGF-β [124]. Its knockout was found to effectively protect mice from bleomycin- and radiation-induced lung fibrosis [125].Collagen type IV alpha 1 and collagen type IV alpha 2 (Col4a1 and Col4a2), encoding pro-alpha 1 and pro-alpha 2 chains, respectively, and presenting in the basement membranes in all organs [126]. It was shown that the expression levels of these genes were increased in the toxic [127] and high fat diet associated [128] liver fibrosis. Moreover, few studies have explored the role of Col4a1 and Col4a2 in pulmonary fibrosis via TGF-β pathway modulation [129,130].

It should be emphasized that only a part of the listed genes have been verified here as probable markers of asthma-driven fibrosis (*Fn1*, *C3*, *C3ar1*, *Cyp2e1*, *Col4a1*, and *Col4a2*); an investigation of the marker function of residuary genes (*Thbs1*, *Ccr3*, *Ccl6*, *Ccl9*, *Tnc*, and *Itgb6*) in pulmonary fibrosis is the subject of future studies.

It is well known that a number of chronic respiratory diseases are accompanied by the development of airway remodeling and pulmonary fibrosis [131,132]. To understand how tightly identified pulmonary fibrosis-related key genes interconnect with fibrosis-accompanied chronic respiratory diseases in humans, their expression levels were deciphered in the samples previously obtained from patients with idiopathic pulmonary fibrosis (IPF), chronic obstructive pulmonary disease (COPD), chronic emphysema (Em), and cystic fibrosis (CF) (Figure 9). It was found that only IPF was characterized by similar expression profile with asthma- and bleomycin-driven lung fibrosis in mice: a range of known fibrosis-related markers, including *COL1A1*, *COL3A1*, *IGF1*, *CXCL12*, *SPP1*, and *MUC5B*, were found to be upregulated in lung tissue but not in peripheral blood of IPF patients (Figure 9). As for COPD, gene expression patterns partially similar to IPF were found in BAL fluid/sputum, but not in lung tissue and peripheral blood. The rest of the analyzed pulmonary disorders were weakly associated with revealed genes that clearly indicated the suitability of the used murine model of asthma-driven fibrosis to clarify some aspects of IPF pathology, but not COPD, CF, or emphysema.

We suppose that observed contradictory enrichment of mentioned disorders by key fibrotic genes (Figure 9) can be explained by differences in their severity: in the case of IPF, the analyzed lung tissues were obtained via diagnostic biopsy at the non-late stage of the disease [133,134], whereas COPD and emphysema samples were mainly collected during thoracic surgery in severe course of mentioned diseases [135,136,137]. Besides this, the sampling site of the listed pulmonary disorders can also affect the obtained results; for example, blood samples and nasal epithelial cells were characterized by almost a complete lack of expression of revealed fibrosis-associated genes, and their expression in the cells isolated from BAL fluid and sputum was significantly different compared to that in lung tissues (Figure 9). Moreover, it should also be noted that IPF and COPD/emphysema are developed on the diverse etiological and pathogenetic background: IPF is based on the impaired wound healing, inducing profibrotic signaling and, as a consequence, lung remodeling [133], and vice versa, COPD/emphysema is based on inflammation, infection, and immune disorders, also eventually leading to pulmonary fibrosis [138], which may cause certain inconsistencies in the gene expression patterns in the lung tissue.

Thus, the identification of a range of the same DEGs in lung tissues of mice with asthma-driven fibrosis and patients with IPF demonstrates a translational bridge between analyzed pathologies. The absence of more pronounced overlap by the revealed key fibrosis-associated genes can be explained by different time scales of the fibrosis development in mice (within a month) and human (years or decades). To test this hypothesis more thoroughly, further evaluation of the expression of identified key genes depending on severity of pulmonary disorders is needed.

### Limitations of the Study

Our study has several limitations to consider. (1) While analysis of public datasets is a versatile and widely available tool, the amount of information regarding type, homogeneity, quality, and processing of the samples varies greatly between each dataset. Despite our efforts to mitigate these factors through a rigorous dataset selection process, using only datasets with the most information about sample processing and with high-powered groups in our analysis, there still may be a certain inconsistency in our results. Additionally, both in our experiments and in most of the public datasets, lung tissue but not isolated cells were analyzed, which may introduce a certain bias into our results. (2) The text mining method used in this study has several challenges to be overcome, and the results of this analysis may be prone to misinterpretation [139]. However, in the present study, the text mining approach was used only to evaluate how widely genes of interest have been investigated in the field of fibrosis without implying their regulatory functions in this disorder. (3) In this study, both in asthma-related bioinformatics analysis and in animal experiments, the model of ovalbumin (OVA)-induced asthma was used. Currently, a rodent house dust mite (HDM)-induced asthma model is starting to replace the model of OVA-driven asthma, since it mimics human asthma more closely [140,141,142]. Although the lack of data from HDM-derived asthma is a limitation of this study, it should be noted that HDM-induced allergic airway inflammation is markedly less reproducible compared to its OVA-induced counterpart since the composition of HDM extracts is slightly standardized, which affects the properties of the model [143]. (4) Only a limited list of revealed key genes associated with asthma and pulmonary fibrosis was validated in vivo. Despite a significant correlation between their expressions measured by cDNA microarray and RT-PCR approaches, which demonstrates high reliability of obtained bioinformatics data, further experimental verification of other identified asthma/fibrosis-associated genes is required.

## 5. Conclusions

According to extensive literature data [144,145], the development and progression of bronchial asthma, one of the most common chronic respiratory diseases, frequently results in the formation of lung fibrosis. Performed complex bioinformatics analysis of murine asthma-related cDNA microarray data and their overlapping with fibrosis-associated genes restored from DisGeNET and transcriptomic profiles of bleomycin-driven pulmonary fibrosis with subsequent validation of the revealed genes on the in vivo model of OVA-induced asthma/fibrosis enabled us identify a range of key genes associated with the transition of acute asthma-specific inflammation to airway remodeling and lung fibrosis, including well-known pro-fibrotic regulators (*Cat*, *Ccl2*, *Ccl4*, *Ccr2*, *Col1a1*, *Cxcl12*, *Igf1*, *Muc5ac/Muc5b*, *Spp1*, *Timp1*) as well as novel genes (*C3*, *C3ar1*, *Col4a1*, *Col4a2*, *Cyp2e1*, *Fn1*, *Thbs1*, *Tyrobp*), which can mediate early fibrotic changes in lungs already at the stage of acute/subacute inflammation. It should also be emphasized that the in vivo model of OVA-induced asthma used in this study clearly demonstrated reliable fibrotization already at the stage of resolution of acute inflammation, but not in the long-term period. Moreover, the validation of genes regulating pulmonary fibrosis of non-allergic etiology (bleomycin-induced lung fibrosis) on asthmatic and asthma-driven fibrotic lungs allows us to identify new promising universal genes (*Col4a1* and *Col4a2*) involved in the development of pulmonary fibrosis regardless of its etiology. The obtained data also demonstrate certain similarities in the expression profiles of nodal fibrotic genes between asthma-driven fibrosis in mice and nascent idiopathic pulmonary fibrosis (IPF) in humans that suggest a tight association of identified genes with the early stages of airway remodeling and clearly show the expediency of their further investigation as predictors and early markers of pulmonary fibrosis.

## Figures and Tables

**Figure 1 biomedicines-10-01017-f001:**
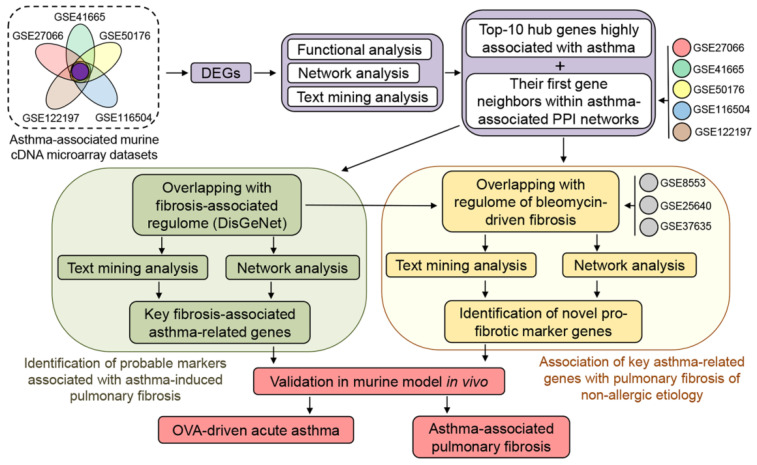
Workflow of the present study.

**Figure 2 biomedicines-10-01017-f002:**
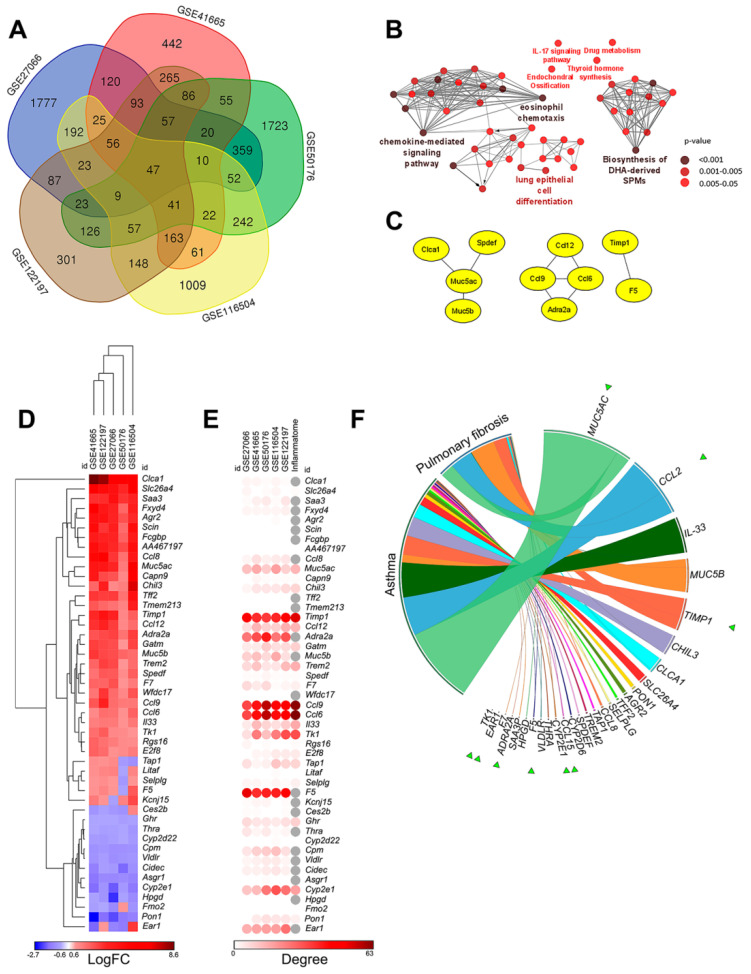
Key genes involved in the development of ovalbumin (OVA)-induced asthma identified by bioinformatics analysis. (**A**) Venn diagram of overlap between differentially expressed genes (DEGs) identified by re-analysis of GSE41665, GSE116504, GSE122197, and GSE27066 datasets of OVA-induced asthma and GSE50176 dataset of nanotube-induced allergic lung inflammation. (**B**) Functional analysis of overlapped DEGs identified in different GSEs. Enrichment for Gene Ontology (biological processes), KEGG, REACTOME, and WikiPathways terms were performed using ClueGo plugin in Cytoscape. The labels of the most significant terms are shown. Color of nodes represents the term enrichment significance. Only pathways with *p* < 0.05 after Bonferroni step-down correction for multiple testing were included in the networks. (**C**) Protein–protein interaction (PPI) network reconstructed for the 47 common DEGs between all asthma-related datasets using STRING database (confidence score ≥ 0.7, maximal number of interactors = 0) in Cytoscape. (**D**) Heat map demonstrating expression levels of DEGs in different GSEs. Heat map construction and hierarchical clustering (Euclidian distances) were performed using Morpheus [38]. LogFC = Log_2_ (fold change). (**E**) Heat map demonstrating interconnection of DEGs in PPI reconstructed for each asthma-related dataset using STRING database (confidence score ≥ 0.7, maximal number of interactors = 0) in Cytoscape. Degree—number of interactions between DEGs and its partners. (**F**) Co-occurrence of identified DEGs with relevant keywords in scientific literature deposited in the MEDLINE database. Analysis was performed using GenClip3 web service. Data were visualized via Circos. Green triangles indicate key genes included in the top-10 interconnected DEGs.

**Figure 3 biomedicines-10-01017-f003:**
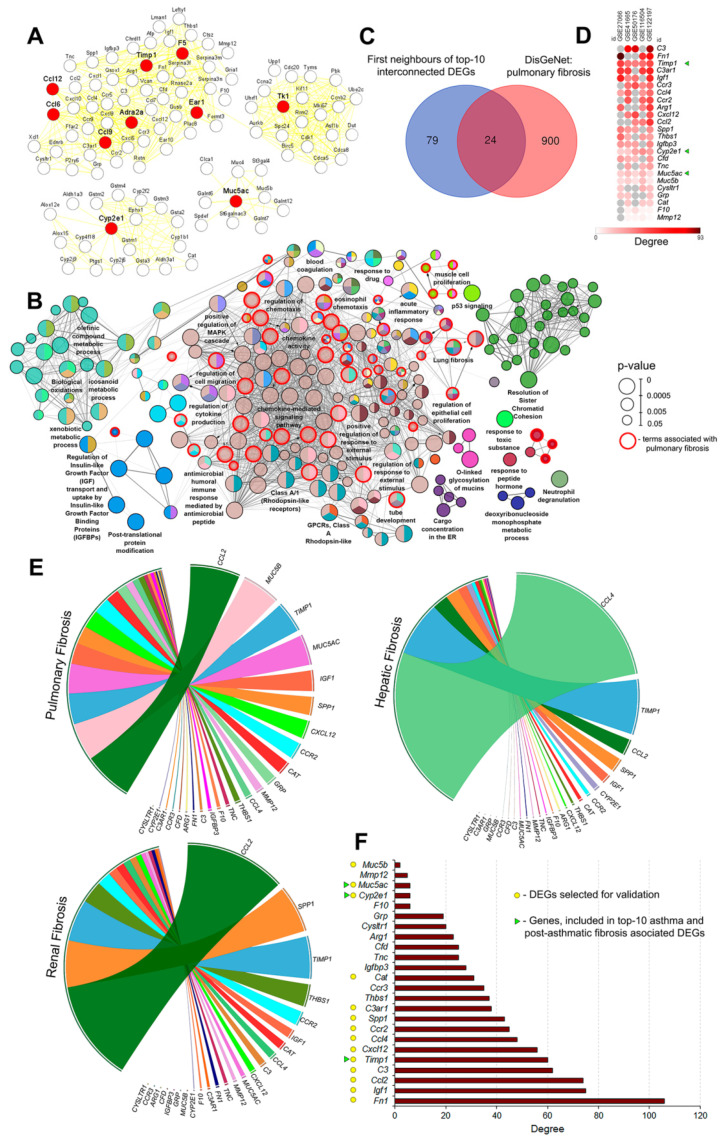
Involvement of key DEGs associated with OVA-induced asthma in the development of pulmonary fibrosis. (**A**) PPI network reconstruction for the top-10 interconnected DEGs with their first neighbors presented in at least three GSEs using STRING database (confidence score ≥ 0.7, maximal number of interactors = 0) in Cytoscape. (**B**) Functional analysis of top-10 interconnected DEGs with their first neighbors presented in at least three GSEs. Enrichment for Gene Ontology (biological processes), KEGG, REACTOME, and WikiPathways terms were performed using ClueGo plugin in Cytoscape. The labels of the most significant terms are shown. Color of nodes represents the term enrichment significance. Only pathways with *p* < 0.05 after Bonferroni step-down correction for multiple testing were included in the networks. Nodes with red boundaries indicate nodes related to fibrosis and lung remodeling. Different gene clusters are marked with different colors automatically. (**C**) Venn diagram demonstrating common genes between top-10 interconnected DEGs with their first neighbors presented in at least three GSEs and genes associated with the development of pulmonary fibrosis, according to the DisGeNET database. (**D**) Heat map showing the interconnection of 24 identified DEGs in PPI networks reconstructed for each OVA dataset using STRING database (confidence score ≥ 0.7, maximal number of interactors = 0) in Cytoscape. Degree—number of interactions between DEGs and its partners. Green triangles indicate key genes included in the top-10 interconnected DEGs. (**E**) Co-occurrence of identified DEGs with keywords associated with the lung, liver, and kidney fibrosis in scientific literature deposited in the MEDLINE database. Analysis was performed using GenClip3 web service. Data are visualized via Circos. (**F**) Degree of 24 identified DEGs in PPI networks reconstructed for genes associated with pulmonary fibrosis, according to DisGeNET, using STRING database in Cytoscape. Degree—number of interactions between DEGs and its partners. Green triangles indicate key genes included in the top-10 interconnected DEGs. Yellow circles indicate DEGs selected for validation.

**Figure 4 biomedicines-10-01017-f004:**
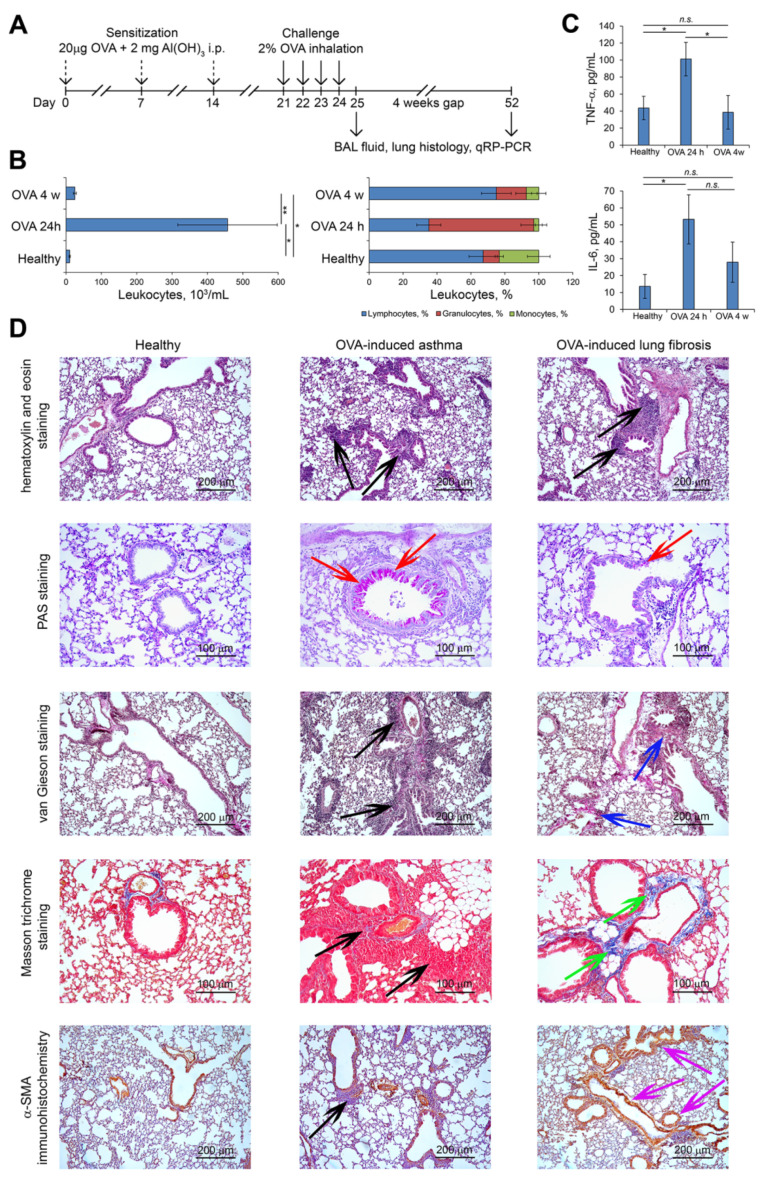
Ovalbumin (OVA)-induced murine model of asthma. (**A**) The experimental setup. BALB/c mice were sensitized by threefold intraperitoneal (i.p.) injection of 20 μg OVA/2 mg aluminum hydroxide (Al(OH)_3_) mixture and then challenged with aerosolized 2% OVA for four consecutive days. Samples for bronchoalveolar (BAL) fluid, histopathological, and qRT-PCR analysis were collected 24 h (day 25) and 4 weeks (day 52) after the last OVA challenge. (**B**) Total (left) and differential (right) leukocyte counts in the BAL fluid of healthy and OVA-challenged mice 24 h and 4 weeks after induction. Total leukocyte counts were estimated using Neubauer chamber. The distribution of leukocyte subpopulations was measured by light microscopy after Romanowsky–Giemsa staining. (**C**) The levels of pro-inflammatory cytokines TNF-α (upper panel) and IL-6 (bottom panel) in BAL fluid measured by ELISA. The data are shown as mean ± standard deviation. The statistical analysis was performed using the two-tailed unpaired *t*-test; * *p* ≤ 0.05, ** *p* ≤ 0.01, n.s., not significant. (**D**) Representative histological images of lung sections of healthy and OVA-challenged mice 24 h and 4 weeks after induction. Original magnification ×100 (hematoxylin and eosin staining, van Gieson staining, α-SMA staining) and ×200 (PAS staining, Masson trichrome staining). The black arrows indicate inflammatory infiltration, the red arrows indicate mucus production, the blue arrows indicate elastic fibers by van Gieson staining, the green arrows indicate collagen fibers by Masson trichrome staining, and the magenta arrows indicate stromal component by α-SMA staining.

**Figure 5 biomedicines-10-01017-f005:**
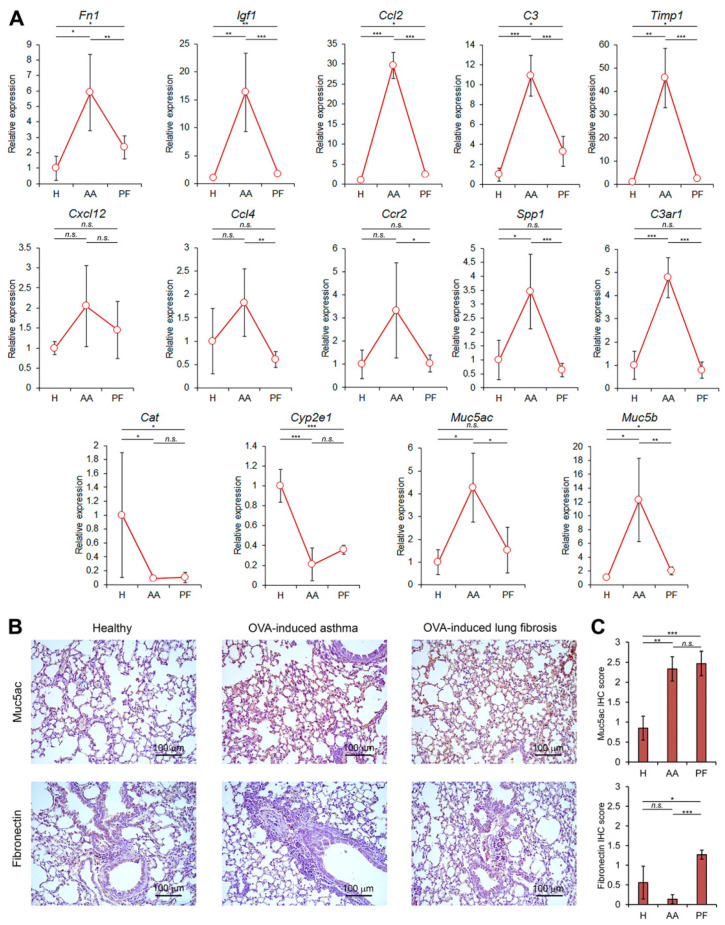
Expression of key genes identified by bioinformatics analysis in the lung tissue of OVA-challenged mice. (**A**) qRT-PCR data for healthy lungs (H), acute asthma (AA, 24 h after the last OVA challenge), and pulmonary fibrosis (PF, 4 weeks after the last OVA challenge). Expression levels were normalized to the expression level of hypoxanthine phosphoribosyltransferase (*HPRT*) (used as the reference gene). Three to five samples from each experimental group were analyzed in triplicate. The data are shown as mean ± standard deviation. The statistical analysis was performed using the two-tailed unpaired *t*-test; * *p* ≤ 0.05, ** *p* ≤ 0.01, *** *p* ≤ 0.001, n.s., not significant. (**B**) Representative immunohistochemical images of lung sections of healthy and OVA-challenged mice 24 h (OVA-induced asthma) and 4 weeks (OVA-induced lung fibrosis) after induction stained with anti-Fibronectin and anti-Muc5ac primary antibodies. Original magnification ×200. (**C**) The intensity of immunohistochemical staining of lung sections was assessed by a semi-quantitative method, where 0 is no expression, 1 is low expression, 2 is moderate expression, and 3 is high expression. The number of samples studied was three for each experimental group. The statistical analysis was performed using the two-tailed unpaired *t*-test; * *p* ≤ 0.05, ** *p* ≤ 0.01, *** *p* ≤ 0.001, n.s., not significant.

**Figure 6 biomedicines-10-01017-f006:**
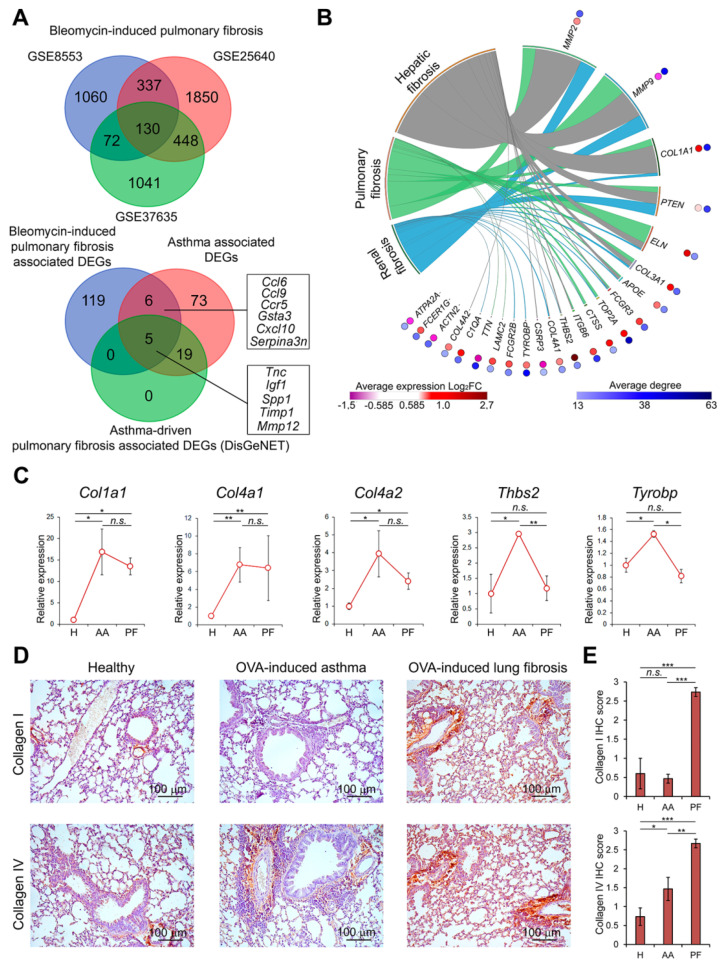
Involvement of genes associated with bleomycin-induced lung fibrosis in the development of asthma and post-asthmatic fibrosis. (**A**) Venn diagram of genes differentially expressed in the lungs of mice with bleomycin-induced pulmonary fibrosis (upper panel). Venn diagram of overlapping genes differentially expressed in the lungs of mice with bleomycin-induced pulmonary fibrosis and top-10 asthma-associated genes with their first neighbors (lower panel). (**B**) Co-occurrence of identified DEGs with keywords associated with pulmonary, hepatic, and renal fibrosis in scientific literature deposited in the MEDLINE database. Analysis was performed using GenClip3 web service. Data were visualized via Circos. Red circles: heat map demonstrating expression levels of DEGs in different GSEs. LogFC = Log_2_ (fold change). Blue circles: heat map demonstrating interconnection of DEGs in PPI reconstructed using STRING database (confidence score ≥ 0.7, maximal number of interactors = 0) in Cytoscape. Degree—number of interactions between DEGs and its partners. (**C**) Expression of fibrosis-associated genes in the lungs of healthy mice and mice with OVA-induced asthma (OVA 24 h) and post-asthmatic fibrosis (OVA 4 w). Relative expression levels were normalized to the expression level of hypoxanthine phosphoribosyltransferase (*HPRT*) (used as the reference gene). Three to five samples from each experimental group were analyzed in triplicate. The data are shown as mean ± standard deviation. (**D**) Representative immunohistochemical images of lung sections of healthy mice and mice with OVA-induced asthma and post-asthmatic fibrosis stained with anti-collagen I and anti-collagen IV primary antibodies. Original magnification ×200. (**E**) The intensity of immunohistochemical staining of lung sections was assessed by a semi-quantitative method, where 0 is no expression, 1 is low expression, 2 is moderate expression, and 3 is high expression. The number of samples studied was three for each experimental group. The data are shown as mean ± standard deviation. The statistical analysis was performed using the two-tailed unpaired *t*-test; * *p* ≤ 0.05, ** *p* ≤ 0.01, *** *p* ≤ 0.001, n.s., not significant.

**Figure 7 biomedicines-10-01017-f007:**
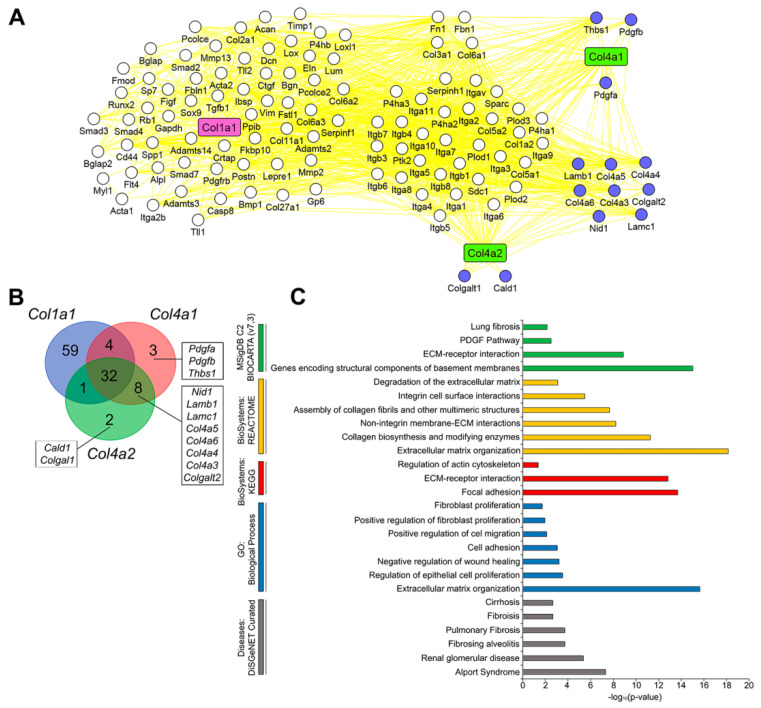
Functional analysis of the *Col4a1* and *Col4a2* and their associated genes. (**A**) PPI of *Col1a1*, *Col4a1*, *Col4a2*, and their first neighbors. Blue nodes mark the genes, associated with either *Col4a1*, *Col4a2*, or both of them. (**B**) Venn diagram, demonstrating the overlap between the first neighbors of *Col1a1*, *Col4a1*, and *Col4a2*. (**C**) Histogram, visualizing functional enrichment analysis of *Col4a1*, *Col4a2*, and their first neighbors. Functional enrichment analysis was performed through the ToppFun module of the ToppGene suite, using the databases MSigDB Biocarta (v7.3), BioSystems: REACTOME and KEGG, Gene Ontology: biological processes and DisGeNET curated database. For all enrichment tools, the input gene set was composed of the same 15 genes and used with default options: significance threshold of 0.05 for adjusted *p*-value, at least two genes from the input list in the enriched category, and the whole genome as the reference background.

**Figure 8 biomedicines-10-01017-f008:**
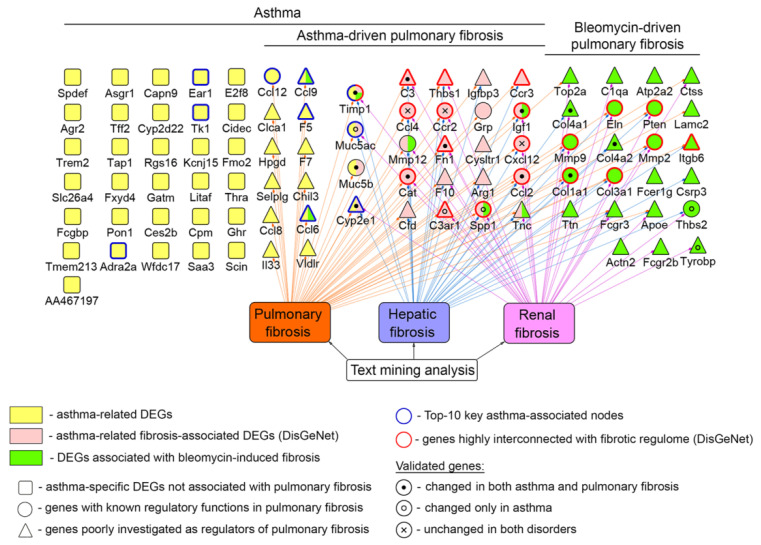
Interconnection of asthma-associated genes with pulmonary fibrosis regulome.

**Figure 9 biomedicines-10-01017-f009:**
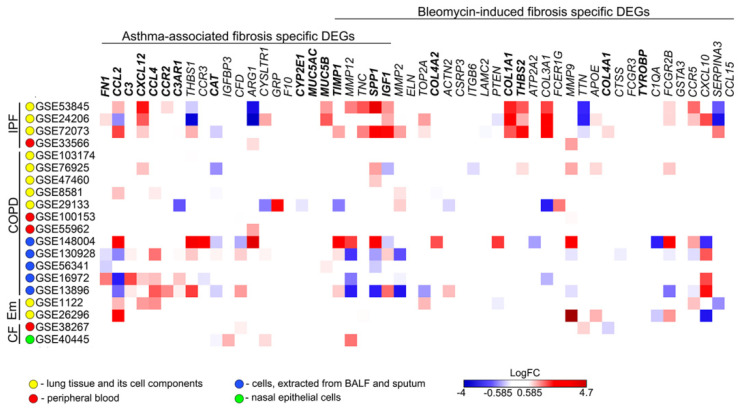
Expression of asthma- and bleomycin-associated pulmonary fibrosis DEGs in human chronic lung pathology. Heat map demonstrating expression levels of DEGs connected with asthma-associated and bleomycin-induced lung fibrosis in human datasets. Heat map construction was performed using Morpheus. LogFC = Log_2_ (fold change). Different types of biological materials were used: lung tissues (GSE53845, GSE24206, GSE72073, GSE8581, GSE103174, GSE76925, GSE47460, and GSE1122), peripheral blood (GSE33566, GSE100153, and GSE28267), alveolar macrophages (GSE130928, GSE16972, and GSE13896), sputum (GSE148004), epithelial cells of small airways (GSE56341), total leukocytes from peripheral blood (GSE55962), alveolar epithelial cells type II (GSE29133), lung myeloid cells (GSE26296), and nasal epithelial cells (GSE40445). IPF, idiopathic pulmonary fibrosis; COPD, chronic obstructive pulmonary disease; Em, chronic emphysema; CF, cystic fibrosis. Bold font denotes genes validated in this study in the murine model of OVA-induced asthma and asthma-driven pulmonary fibrosis.

**Table 1 biomedicines-10-01017-t001:** The expression levels of genes identified as molecular markers of asthma and lung fibrosis by bioinformatics approaches.

Gene ID	Description	Microarray Data Fold Change ^1^					Experimental Fold Change ^2^	
		GSE27066 ^#^	GSE41665 ^#^	GSE50176 ^#^	GSE116504 ^#^	GSE122197 ^#^	Asthma 3	Fibrosis 4
*Timp1*	Tissue Inhibitor of Matrix Metalloproteinase 1	9.4	8.2	1.9	3.6	17.9	45.7 *	2.3 *
*Ccl2*	C-C Motif Chemokine Ligand 2	n/d	n/d	2.4	3.2	7.2	29.5 ***	2.3 *
*Igf1*	Insulin-like growth factor 1	3.2	2.5	1.6	n/d	7.6	16.3 *	1.7 *
*Muc5b*	Mucin 5b	4.7	6.6	3.2	2.9	5.6	12.2 *	2 *
*C3*	Complement component 3	1.4	1.6	1.9	n/d	3.2	10.8 ***	3.3 *
*Fn1*	Fibronectin 1	1.9	n/d	n/d	3.3	2.3	5.9 *	2.3 *
*Muc5ac*	Mucin 5ac	3.1	3.9	1.2	6.7	3.9	5.7 *	1.5
*C3ar1*	Complement C3a receptor 1	3.5	4.8	n/d	1.5	3.6	4.7 ***	−1.6
*Spp1*	Secreted phosphoprotein 1	1.9	3.3	1.5	n/d	5.8	3.4 *	−1.6
*Ccr2*	C-C Motif Chemokine receptor 2	n/d	2.3	n/d	2.1	3	3.3	1
*Cxcl12*	C-X-C Motif Chemokine Ligand 12	n/d	2.1	−1.7	n/d	−1.5	2	1.5
*Ccl4*	C-C Motif Chemokine Ligand 4	1.5	1.2	n/d	−1.5	1.6	1.8	−1.6
*Cyp2e1*	Cytochrome P450 Family 2 Subfamily E Member 1	−3.5	−3	−1.8	−2.8	−1.9	−4.8 ***	−2.8 ***
*Cat*	Catalase	−1.5	−1.5	n/d	−1.7	−1.5	−11.2	−9.6 *

^1^ Gene expression data were obtained by the comparison of lung tissues of asthmatic mice with lung tissues of healthy mice. ^#^ GSE27066 and GSE41665—platform: GPL1261 [Mouse430_2] Affymetrix Mouse Genome 430 2.0 Array; induction stimuli: OVA; source: lung tissue. GSE50176—platform: GLP13912 Agilent-028005 SurePrint G3 Mouse GE 8 × 60 K Microarray; induction stimuli: rod shaped carbon nanotubes; source: lung tissue. GSE116504—platform: GPL21164 Agilent-074809 SurePrint G3 Mouse GE v2 8 × 60 K Microarray; induction stimuli: OVA; source: lung tissue. GSE122197—platform: GPL21810 Agilent-074809 SurePrint G3 Mouse GE v2 8 × 60 K Microarray; induction stimuli: OVA; source: lung tissue. ^2^ Expression levels in experimental groups were normalized to the expression levels in healthy mice. Six samples from each experimental group were analyzed in triplicates. ^3^ Lungs of mice 24 h after the last OVA induction (asthmatic lungs). ^4^ Lungs of mice 4 weeks after the last OVA induction (fibrotic lungs). n/d, no data. Statistical analysis was performed using the two-tailed unpaired *t*-test; * *p* ≤ 0.05, *** *p* ≤ 0.001, n.s., not significant compared to healthy controls.

## Data Availability

The data presented in this study are openly available on the Gene Expression Omnibus database; reference numbers GSE27066, GSE41665, GSE116504, GSE122197, GSE50176, GSE8553, GSE25640, GSE37635, GSE53845, GSE24206, GSE72073, GSE33566, GSE103174, GSE76925, GSE47460, GSE8581, GSE29133, GSE100153, GSE55962, GSE148004, GSE130928, GSE56341, GSE16972, GSE13896, GSE1122, GSE26296, GSE38267, and GSE40445.

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
