# Peer review of "Asthma and Post-Asthmatic Fibrosis: A Search for New Promising Molecular Markers of Transition from Acute Inflammation to Pulmonary Fibrosis"

_biomedicines, 2022, doi:10.3390/biomedicines10051017_

Round 1

Reviewer 1 Report

This manuscript aims to reveal the molecular mechanisms involved in the asthma progression and lung fibrosis. This manuscript has significant limitations which are detailed in the comments below.

  1. In general the manuscript is difficult to read. Abstract, introduction and discussion need to be largely improved. Introduction and discussion need to be rewritten: they are too long and do not focus on the essential concepts of the problem.
  2. The aim of the study, which is not completely clear, should be better explained.
  3. Methods is not clear. I suggest that the authors rewrite the methods, transferring some parts to the online data supplement. Authors should also mention the statistics program used.
  4. It is not clear how the findings in mouse could correlate to those in humans, especially considering that the bleomycin model of fibrosis does not reflect human fibrosis. The authors should stress this aspect in the discussion.
  5. I would advise the authors to reduce the results as well, trying to focus on the most important ones.
  6. I would advise the authors to be very cautious in their conclusions especially in the first sentence where they state that the COPD “inevitably results in the formation of lung fibrosis”. The authors should report references that support this statement.

Author Response

Reviewer #1.

This manuscript aims to reveal the molecular mechanisms involved in the asthma progression and lung fibrosis. This manuscript has significant limitations which are detailed in the comments below.

  1. In general the manuscript is difficult to read. Abstract, introduction and discussion need to be largely improved. Introduction and discussion need to be rewritten: they are too long and do not focus on the essential concepts of the problem.

Dear Reviewer #1,

Thank you so much for the valuable comments and suggestions that helped us to improve the manuscript. We revised and modified the manuscript according to your comments (revised parts are marked by red). According to your recommendation, English editing of our article was performed by MDPI Language Editing Service.

Abstract was modified by specification the list of genes validated in our study (please, see lines 15-18). The Introduction section was corrected by shortening the paragraphs concerning the description of ECM (please, see lines 34-41) and by transferring the description of the murine asthma model to the Results section (please, see lines 410-418). According to your comments, in the Discussion section we tried enhancing the focus of readers’ attention only on the main findings of the study, namely on the revealed novel marker genes of asthma-driven fibrosis and their association with pulmonary fibrosis in humans. To do this, the paragraphs describing used murine asthma model and pro-fibrotic functions of type IV collagens were transferred from the Discussion to the Results section (please, see lines 481-489, 658-693). Moreover, to make the bioinformatics analysis described in the manuscript more understandable, the workflow of the study was added to the Introduction section (please, see Scheme 1).

  1. The aim of the study, which is not completely clear, should be better explained.

Corrected. The paragraph concerning the aim of our study was modified to make it clearer and better explained (please, see lines 57-66).

  1. Methods is not clear. I suggest that the authors rewrite the methods, transferring some parts to the online data supplement. Authors should also mention the statistics program used.

Corrected. We shortened the Materials and Methods section by transferring Table containing list of primers used in the study as well as methods describing functional enrichment analysis, PPI networks reconstruction and ELISA assay to the Supplementary Materials (please, see Supplementary Materials, Table S2 and lines 114-119, 158-161). We clarified the statistical methods and programs used in our study in more detail (please, see lines 216-225).

  1. It is not clear how the findings in mouse could correlate to those in humans, especially considering that the bleomycin model of fibrosis does not reflect human fibrosis. The authors should stress this aspect in the discussion.

To understand how tightly pulmonary fibrosis-related key genes identified in mice interconnect with fibrosis-accompanied chronic respiratory diseases in human, we compared their expression levels with that in the samples obtained from patients with chronic lung pathology accompanied with pulmonary fibrosis: idiopathic pulmonary fibrosis (IPF), chronic obstructive pulmonary disease (COPD), chronic emphysema and cystic fibrosis (CF) (please, see Figure 8). It was found that only IPF in human is characterized by similar expression profile with asthma- and bleomycin-driven lung fibrosis in mice: a range of known fibrosis-related master regulators, including COL1A1, COL3A1, IGF1, CXCL12, SPP1 and MUC5B, were found to be up-regulated in lung tissue of IPF patients (please, see Figure 8). The rest of the analyzed human pulmonary disorders were weakly associated with revealed genes that clearly indicated suitability of used murine model of asthma-driven fibrosis to clarify some aspects of IPF pathology, but not COPD, CF or emphysema. Thus, identification of a range of the same DEGs in lung tissues of mice with asthma-driven fibrosis and patients with IPF demonstrates a translational bridge between analyzed pathologies.

These moments were highlighted in the Discussion section (please, see lines 820-870).

  1. I would advise the authors to reduce the results as well, trying to focus on the most important ones.

Unfortunately, we were unable to markedly reduce the Results section without losing the clarity of the description of our results. Besides, the other two reviewers insisted on expanding the Results section to make it more detailed and understandable to reader. However, to attempt to reduce text in the Results section, descriptions of disorders with which identified hub genes can be associated were significantly shortened (please, see lines 285-293). Moreover, we hope that the workflow of our analysis depicted in Scheme 1 will be helpful for readers to more thoroughly understand the tasks of each step of performed bioinformatics analysis and, thus, not to be confused by the obtained multi-parameter results. The references on Scheme 1 were introduced in the Results section (please, see lines 373 and 587).

  1. I would advise the authors to be very cautious in their conclusions especially in the first sentence where they state that the COPD “inevitably results in the formation of lung fibrosis”. The authors should report references that support this statement.

Corrected. We have softened our statement regarding the inevitability of pulmonary fibrosis in the asthma development and progression and supported our conclusions about high incidence of fibrosis in this pathology by the relevant references (please, see lines 898-900).

Reviewer 2 Report

Savin et. al. explore changes in gene expression in asthma and post-asthmatic fibrosis using bioinformatics approaches and a rodent model of OVA-induced asthma. The manuscript is quite comprehensive and generally well developed. Some weaknesses relate to the animal model being underpowered and that findings are largely descriptive (e.g. functional significance of specific genes is implied, not tested; biomarkers are not associated with disease severity or specific outcomes).

Specific points:

  • I would suggest removing the word “regulators” from the Title. While the identified asthma/fibrosis-related genes are clearly “markers” of the disease and its progression, their role in “regulating” the transition from acute inflammation to pulmonary fibrosis (e.g. establish causality) is not tested here.
  • While ideas are generally well formulated, the manuscript requires some English language editing focused on grammar and punctuation as exemplified by the Title (… a search for “a” new promising molecular markers…; plural/delete “a”) or the first sentence of the Abstract (…chronization of which “led” to…; present tense/”leads to”).
  • Abstract Ln13-17. The authors state that bioinformatics data were validated in an animal model and follow this up with a list of genes associated with disease. However, only a fraction of these genes were actually validated experimentally.
  • The authors are correct in that the OVA model is a historically widely used animal model for asthma. However, as asthmatics rarely inhale egg whites, rodent models of house-dust mite-induced asthma are increasingly replacing OVA-models given their close parallels to human patients. To be clear, I am not asking the authors to perform new animal experiments. However, they may consider incorporating datasets from house-dust mite models in their bioinformatic analyses and/or just state the lack of these data/considerations as potential limitations of the study.
  • Section 2.1./Ln100-120: Please include the tissue/material used in the description of all datasets (e.g. “lung”, “BAL”, “blood”).
  • Ln 115-120: pleased indicate/clarify the species (e.g. “human”) for all datasets listed.
  • Ln 402-411: The text mining approach seems like a crude tool and prone to misinterpretations as it appears to utilize words detached from their context. For example, how would this search interpret the last sentence of this paragraph (Ln 410/411) which states that “…muc5B, Cyp2e1, and C3 are specific to pulmonary, hepatic, and renal fibrosis, respectively.” Would it interpret the unique association between genes and tissues correctly, or associated any of the three genes with any of the three tissues? Should this be discussed as a potential limitation? Would a text stating that expression of gene X is “unaltered” in fibrosis, lead to the analysis assuming that gene X is “associated” with fibrosis?
  • N numbers are frequently missing. Please state the number of animals used per experimental group and assay for all animal experiments (e.g. Methods section and figure legends).
  • Figure 3C/4B; the authors should consider a blinded scoring/quantification of the injury seen in mouse lungs, rather than showing representative images.
  • Ln470: authors state that BAL leukocytes in the subacute group do not return to healthy levels, but no statistical analyses are presented.
  • 3A/Table 2/Fig. 5C: with n=3 biological replicates, data from animal experiments shown in these figures/tables seem severely underpowered.
  • The value of Table 2 would be enhanced by adding statistics (e.g. symbols indicating presence/absence of statistical significance).
  • Ln 554/Fig. 4B: It is unclear how the authors can claim that expression of various proteins (e.g. Muc5ac) are “significantly” changed, if no quantification/statistics are provided.
  • Ln 561/562: Time lag between mRNA and protein expression seems unlikely in a weeks-long disease model.
  • Ln 616-618: The authors should consider that not all genes changed in disease states can be “master regulators”.
  • 5D, please provide quantification.
  • Ln677: The authors state that early fibrotic changes are detected in the absence of repeated OVA cycles (Fig. 3). This is confusing to me as the mice at days 25 or 52 (Fig. 3A) have already undergone multiple cycles of OVA.
  • Ln816-828/Fig. 8: Given the cell-type specific expression of various genes, the authors correctly point out that one should not expect them to be detected similarly in the various samples (e.g. BAL cells versus nasal epithelial cells versus peripheral blood). So what is the point then? How does the source of the sample affect the value of various genes as biomarkers?
  • LN115-120/LnLn816-828: Maybe I missed something here, but why are rodent asthma models not first compared to human asthma data sets? Why compare rodent asthma models to any other chronic lung disease except for asthma?    
  • Ln872: Please revise “regulating” as this would imply demonstrated causality. All that is demonstrated here is coincidence (correlation in time).
  • Ln 871-876: The text implies that a long list of genes was associated with fibrosis using bioinformatics approaches and subsequently confirmed in an in vivo model. This is not the case as only a small subset of genes was tested/confirmed in the mouse model.

Author Response

Reviewer #2.

Savin et. al. explore changes in gene expression in asthma and post-asthmatic fibrosis using bioinformatics approaches and a rodent model of OVA-induced asthma. The manuscript is quite comprehensive and generally well developed. Some weaknesses relate to the animal model being underpowered and that findings are largely descriptive (e.g. functional significance of specific genes is implied, not tested; biomarkers are not associated with disease severity or specific outcomes).

Specific points:

  • I would suggest removing the word “regulators” from the Title. While the identified asthma/fibrosis-related genes are clearly “markers” of the disease and its progression, their role in “regulating” the transition from acute inflammation to pulmonary fibrosis (e.g. establish causality) is not tested here.

Dear Reviewer #2,

We are very grateful to you for the valuable comments and suggestions that helped us to improve the manuscript. We revised and modified the manuscript according to your comments (revised parts are marked by red).

We agree with your criticism of incorrect usage of phrases describing the regulatory functions of revealed key genes; this is our omission. Indeed, in the current work, we did not use knockout/silencing approaches or selective inhibitors to check the involvement of identified genes in the regulation of asthma/fibrosis progression. To address this problem, the word “regulators” was removed from the Title (please, see lines 2-4) and, moreover, all phrases about key genes as “master regulators” were carefully corrected throughout the text by replacing them with phrases about the “marker” function of key genes (please, see lines 68, 369, 535, 581, 625, 635, 708, 717, 740, 770, 814-817, 828) or by using probabilistic phrases, such as “probable regulatory function” (line 591), “possible involvement in the regulation of” (lines 692-693), “genes associated with” (lines 231, 323, 325, 393, 492, 584, 596, 709, 759).

  • While ideas are generally well formulated, the manuscript requires some English language editing focused on grammar and punctuation as exemplified by the Title (… a search for “a” new promising molecular markers…; plural/delete “a”) or the first sentence of the Abstract (…chronization of which “led” to…; present tense/”leads to”).

Corrected. The manuscript was copyedited by the professional Language Editing Service of MDPI. The certificate of English editing of our article is attached.

  • Abstract Ln13-17. The authors state that bioinformatics data were validated in an animal model and follow this up with a list of genes associated with disease. However, only a fraction of these genes were actually validated experimentally.

Corrected. In the Abstract and Conclusions sections, we mentioned only experimentally validated genes (please, see lines 15-18, 906-908). In the Discussion section, the list of novel probable markers of asthma-associated fibrosis identified in our work contains both verified and unverified key genes (lines 772-813); however, lower we added a paragraph describing which genes from this list were actually validated in vivo and which genes should be validated in the future studies (lines 814-817).

  • The authors are correct in that the OVA model is a historically widely used animal model for asthma. However, as asthmatics rarely inhale egg whites, rodent models of house-dust mite-induced asthma are increasingly replacing OVA-models given their close parallels to human patients. To be clear, I am not asking the authors to perform new animal experiments. However, they may consider incorporating datasets from house-dust mite models in their bioinformatic analyses and/or just state the lack of these data/considerations as potential limitations of the study.

Indeed, rodent models of house dust mite-induced asthma are increasingly replacing OVA-induced models. However, OVA-induced asthma model used in our work fully demonstrates the main characteristics of asthma-driven lung injury, including inflammation, infiltration of eosinophils, production of Th2 cytokines, an increase in serum IgE and airway hyper-responsiveness – clinical and immunologic reactions, which should mimic human pathology [1]. Moreover, OVA challenge models have been long-established worldwide for the preclinical assessment of potential therapeutic agents for asthma [2] and appear to be a more accessible, less complicated and easily reproducible in different laboratory settings. So we considered this model sufficient to gain insight into the molecular mechanisms of asthma development and progression. However, information about the restrictions associated with the usage of OVA-induced asthma model was added to the paragraph about study limitations at the end of the Discussion section (please, see lines 884-891).

  • Section 2.1./Ln100-120: Please include the tissue/material used in the description of all datasets (e.g. “lung”, “BAL”, “blood”).

Corrected. We included information about the used tissue/material in the description of all microarray datasets; data are presented in Table S1.

  • Ln 115-120: pleased indicate/clarify the species (e.g. “human”) for all datasets listed.

Corrected. The information about the species (“human” or “mouse”) of all used microarray datasets was included in the main manuscript (please, see lines 84-85, 91-92, 98) and in Table S1.

  • Ln 402-411: The text mining approach seems like a crude tool and prone to misinterpretations as it appears to utilize words detached from their context. For example, how would this search interpret the last sentence of this paragraph (Ln 410/411) which states that “…muc5B, Cyp2e1, and C3 are specific to pulmonary, hepatic, and renal fibrosis, respectively.” Would it interpret the unique association between genes and tissues correctly, or associated any of the three genes with any of the three tissues? Should this be discussed as a potential limitation? Would a text stating that expression of gene X is “unaltered” in fibrosis, lead to the analysis assuming that gene X is “associated” with fibrosis?

We agree that text mining approach used in the study is a rather crude tool for finding connections between scientific terms, since any mention of a certain gene in relation with certain pathology, even if the gene expression does not change, will be considered as an established link. The mentioned problem is one of the challenges to be overcome to avoid misinterpretation of the results of this analysis [3]. However, in this study, text mining approach was used only to demonstrate a co-occurrence of genes and disorders of interest in abstracts of scientific articles without implying of functional interconnections between them (in other words, using text mining, we tried to understand how thoroughly evaluated genes were investigated in the field of fibrosis). Explanations about issues of text mining approach were added to the study limitations paragraph at the end of the Discussion section (please, see lines 880-884).

  • N numbers are frequently missing. Please state the number of animals used per experimental group and assay for all animal experiments (e.g. Methods section and figure legends).

Corrected. The information concerning the numbers of mice in each experimental group for all animal assays was included in the Materials and Methods section (please, see lines 146-147, 157, 170-171, 185, 214, 505-506, 614), but not in the figure legends to avoid their excessive cluttering and complication.

  • Figure 3C/4B; the authors should consider a blinded scoring/quantification of the injury seen in mouse lungs, rather than showing representative images.

Corrected. The assessment of the intensity of immunohistochemical staining of lung sections by semi-quantitative method was performed, and the following scoring scale was used: 0 – no expression, 1 – low expression, 2 – moderate expression, 3 – high expression. The quantification was performed at a magnification ×200 in 5-6 test fields for each lung sample; the number of samples studied was three for each experimental group. It should be noted that we used aforementioned scoring only in the case of IHC analysis of protein levels of genes selected for validation. Data were included in the main manuscript, Figures and Materials and Methods section (please, see Figures 4C and 5E and lines 181-186, 510-515, 618-623).

  • Ln470: authors state that BAL leukocytes in the subacute group do not return to healthy levels, but no statistical analyses are presented.

Corrected. Statistical analysis of leukocyte number in BAL fluid was performed and included in the Figure 3B. In both OVA-challenged groups the differences were statistically significant from healthy animals: OVA-challenged mice 24 h after induction p=0.028763, OVA challenged mice 4 weeks after induction p=0.026666 compared to healthy control.

  • 3A/Table 2/Fig. 5C: with n=3 biological replicates, data from animal experiments shown in these figures/tables seem severely underpowered.

Corrected. We have included information concerning the number of mice in each experimental group for all animal assays in the Materials and Methods section (please, see lines 146-147, 157, 170-171, 185, 505-506, 614). It should be noted that the number of mice in each experimental group was as follows: animal experiment, histological and histochemical analysis – nine; BAL fluid analysis (leukocytes, ELISA) – five; immunohistochemical study – three; qRT-PCR – from three to five samples analysed in triplicate. Thus, the presented number of animals seems to be sufficient for statistical analysis and identification of significant differences between the studied groups.

  • The value of Table 2 would be enhanced by adding statistics (e.g. symbols indicating presence/absence of statistical significance).

Corrected. Symbols indicating statistical significance were added in Table 2.

  • Ln 554/Fig. 4B: It is unclear how the authors can claim that expression of various proteins (e.g. Muc5ac) are “significantly” changed, if no quantification/statistics are provided.

Corrected. Semi-quantative evaluation of IHC staining intensity of lung sections with subsequent statistical analysis was performed, that can indicate the significance of changes in the expression levels of studied proteins. Obtained data were introduced in Figure and the Materials and Methods section (please, see Figure 4C and lines 181-186).

  • Ln 561/562: Time lag between mRNA and protein expression seems unlikely in a weeks-long disease model.

We agree that an explanation regarding the discrepancy between the mRNA expression level and protein level of Fn1 is unlikely in the context of our experiment. We based our assumption on the fact, that such diversity in mRNA and protein levels of Fn1 has already been demonstrated by Liu et al. on a rat model of CCL4-induced liver fibrosis [4]. However, the correlation between mRNA and protein levels in cells and tissues is a widely debated topic with no clear answer, since there is still no precise understating of the mechanisms behind the potential difference in the mRNA and protein expression [5]. Now the consensus is that while mRNA and protein expression levels are generally well correlated in the steady state and less so in the disease state, there are numerous biological mechanisms, which decouple protein levels from mRNA expression, such as the variance of translation rates regulated by mRNA sequence [6], translation rates modulation [7] and modulation of protein half-life [8], which should be considered when predicting protein levels on the basis of mRNA expression levels [9–15]. Therefore, further investigations to discern reasons behind the discrepancy between the mRNA and protein expression levels of Fn1 are necessary. We added this discussion to the manuscript (please, see lines 561-576).

  • Ln 616-618: The authors should consider that not all genes changed in disease states can be “master regulators”.

We fully agree that not all genes changing their expression in the pathological states should be considered as “master-regulators” and/or perform a regulatory function in general. Given this fact and the absence of experiments confirming the regulatory role of revealed key genes in our work, we re-written the parts of the article regarding the master regulator functions of identified DEGs to imply only possible marker functions of these DEGs, not regulating ones (please, see lines 68, 231, 323, 325, 369, 377, 393, 396, 403, 492, 535, 581, 584, 591, 594, 596, 625, 635, 708, 709, 717, 735, 740, 745, 759, 769, 770, 828, 898, 905, 908). We hope that our findings will be helpful for other research groups studying the mechanisms of pulmonary fibrosis development, which can verify the regulatory functions of key genes revealed in our work experimentally (for instance, using siRNA/shRNA/CRISPR-Cas9 techniques or selective inhibitors).

  • 5D, please provide quantification.

Corrected. Semi-quantative evaluation of IHC staining intensity of lung sections with subsequent statistical analysis was performed. Obtained data were added to Figure 5, Materials and Methods section (please, see Figure 5E and lines 181-186).

  • Ln677: The authors state that early fibrotic changes are detected in the absence of repeated OVA cycles (Fig. 3). This is confusing to me as the mice at days 25 or 52 (Fig. 3A) have already undergone multiple cycles of OVA.

We agree that the description of the performed experiment may be confusing to some extent. Generally, acute asthma is induced by OVA/Al(OH)3 sensitization with a subsequent single cycle of OVA challenges consisting of several OVA inhalations [16]. Further extension of the number of OVA challenge cycles, also consisting of several OVA inhalations, led to the development of chronic changes in the lungs, such as airway remodeling and lung fibrosis [17,18]. In our experiments, only single OVA challenge cycle was used (please, see Figure 3A and Materials and Methods section), while the conventional chronic asthma models employ at least 2 or more such cycles, as seen in [19,20].

  • Ln816-828/Fig. 8: Given the cell-type specific expression of various genes, the authors correctly point out that one should not expect them to be detected similarly in the various samples (e.g. BAL cells versus nasal epithelial cells versus peripheral blood). So what is the point then? How does the source of the sample affect the value of various genes as biomarkers?

As it was find out by the bioinformatics analysis of datasets obtained from patients with chronic lung pathology, the source of samples can noticeably affect study results: the gene expression patters in the lung tissue, peripheral blood, cells extracted from BAL fluid and nasal epithelial cells were completely diverse. The data obtained from murine models were consistent with those obtained from humans only in the case of lung tissue used as a sample source. Based on this, we concluded that the identified genes can be detected and used as markers for their evaluation in the lung tissue, however, the question of their evaluation in peripheral blood and BAL fluid remains open and requires further investigations. Disclosure about sample source is included to the limitations paragraph at the end of the Discussion section (please, see lines 872-880).

  • LN115-120/LnLn816-828: Maybe I missed something here, but why are rodent asthma models not first compared to human asthma data sets? Why compare rodent asthma models to any other chronic lung disease except for asthma?

We agree that it is logical to explore the expression levels of revealed murine-specific asthma/fibrosis-associated genes in human asthma microarray datasets. However, given that the main aim of the current study was to find key marker genes of fibrotic remodeling in lungs, evaluation of their expression in human asthma datasets can be unproductive: compared with datasets of idiopathic pulmonary fibrosis, COPD, emphysema and cystic fibrosis selected by us for a translational meta-analysis (Figure 8) and characterized by fibrosis in lung tissue, the information about the fibrotic status of lung tissue of asthmatic patients sampled for cDNA microarray analysis (datasets collected in GEO) is absent. Nevertheless, we analyzed the following human asthma datasets:

  • GSE143303 (17 endobronchial biopsy samples of non-smoker patients with severe eosinophilic asthma, 11 healthy non-smoker controls);
  • GSE179156 (38 endobrnochial biopsy samples of asthma patients, 29 healthy non-smoker controls, publication using this dataset [21]);
  • GSE130499 ( 44 bronchial epithelial cells samples from severe asthma patients, 38 bronchial epithelial cells of healthy controls [22]);
  • GSE64913 (13 epithelial brushing samples from central airways of severe asthma patients, 23 epithelial brushing samples from central airways of healthy controls [23]);
  • GSE110551 (39 whole blood samples from non-obese asthma patients, 39 whole blood samples from non-obese healthy patients [24]).

The results of this analysis are displayed in Figure below. It can be seen that gene expression patterns in human asthma poorly coincide with DEGs specific for both asthma-associated and bleomycin-induced lung fibrosis in mice. We hypothesize that this discrepancy can be related to the absence of fibrotic changes in the lung samples of patients with asthma. Unfortunately, the testing of this hypothesis is difficult since the data whether patients with asthma have pulmonary fibrosis or not is absent in both the GEO dataset information and listed references. For these reasons, we have decided to not include these data in the manuscript.

Figure. Expression of murine asthma- and bleomycin-associated pulmonary fibrosis DEGs in human asthma. Heat map demonstrating expression levels of DEGs connected with murine asthma-associated and bleomycin-induced lung fibrosis in human asthma datasets. Bold font denotes genes validated in this study in the murine model of OVA-induced asthma and asthma-driven pulmonary fibrosis.

  • Ln872: Please revise “regulating” as this would imply demonstrated causality. All that is demonstrated here is coincidence (correlation in time).

Corrected. All phrases about the regulatory function of genes were replaced throughout the manuscript with phrases describing their (i) <marker> function, (ii) <association> with disorder, or (iii) <probable> involvement in disorder development. Please, see lines 20, 68, 231, 318, 323, 324, 393, 396, 403, 492, 535, 581, 583, 584, 591, 594, 596, 635, 708, 709, 717, 735, 745, 759, 769, 770, 828, 904, 908.

  • Ln 871-876: The text implies that a long list of genes was associated with fibrosis using bioinformatics approaches and subsequently confirmed in an in vivo model. This is not the case as only a small subset of genes was tested/confirmed in the mouse model.

Corrected. We agree that the mentioned fact (identification of a wide range of marker gene candidates and validation of their restricted part only) may raise questions. Indeed, our bioinformatics analysis revealed a panel of genes associated with asthma and asthma- / bleomycin-driven fibrosis, which were compiled in the final network depicted in Figure 7; however, only 19 genes from this panel were evaluated by RT-PCR (Figure 7, nodes marked by dots or crosses). Given that the majority of validated genes were differentially expressed in asthmatic and fibrotic lungs (Figures 4A and 5C), as expected by microarray analysis, we decided that our bioinformatics data is quite reliable and, thus, we can consider identified key genes as probable (!) marker genes associated with evaluated disorder. This algorithm is common for cDNA microarray analysis, which usually requires a verification step, where the expression of only a distinct number of revealed genes of interest (for instance, hub genes) was quantified by RT-PCR (usually, 3–10 genes) [25–27]. And if the expression of the majority of verified genes is similar in both microarray and RT-PCR analysis, authors can conclude that the obtained in silico data indeed reflect real biological process/effect, which provides the opportunity to draw conclusions based on all revealed genes of interest but not only on validated ones.

To not mislead readers, (a) in the Abstract and Conclusions sections, only key genes verified in the murine model were described (please, see lines 15-18, 906-908); (b) in the Discussion section, in the paragraph describing novel potential markers of asthma-associated lung fibrosis we clearly indicated which genes were verified in the model of asthma-driven fibrosis and which were not (please, see lines 814-817); (c) disclosure about the limited subset of genes selected for validation was added to the paragraph devoted to the limitations of the study (please, see lines 892-896).

References

  1. Alessandrini F, Musiol S, Schneider E, Blanco-Pérez F, Albrecht M. Mimicking Antigen-Driven Asthma in Rodent Models—How Close Can We Get? Front Immunol [Internet]. Frontiers Media SA; 2020 [cited 2022 Apr 7];11:575936. Available from: /pmc/articles/PMC7555606/
  2. Azman S, Sekar M, Bonam SR, Gan SH, Wahidin S, Lum PT, et al. Traditional Medicinal Plants Conferring Protection Against Ovalbumin-Induced Asthma in Experimental Animals: A Review. J Asthma Allergy [Internet]. Dove Press; 2021 [cited 2022 Apr 7];14:641. Available from: /pmc/articles/PMC8214026/
  3. Yang Y, Adelstein SJ, Kassis AI. Target discovery from data mining approaches. Drug Discov Today. Elsevier Current Trends; 2009;14:147–54.
  4. Liu XY, Liu RX, Hou F, Cui LJ, Li CY, Chi C, et al. Fibronectin expression is critical for liver fibrogenesis in vivo and in vitro. Mol Med Rep. 2016;14:3669–75.
  5. Lahtvee PJ, Sánchez BJ, Smialowska A, Kasvandik S, Elsemman IE, Gatto F, et al. Absolute Quantification of Protein and mRNA Abundances Demonstrate Variability in Gene-Specific Translation Efficiency in Yeast. Cell Syst. Cell Press; 2017;4:495-504.e5.
  6. Wethmar K, Smink JJ, Leutz A. Upstream open reading frames: Molecular switches in (patho)physiology. BioEssays. John Wiley & Sons, Ltd; 2010. p. 885–93.
  7. Barrett LW, Fletcher S, Wilton SD. Regulation of eukaryotic gene expression by the untranslated gene regions and other non-coding elements. Cell Mol Life Sci. Springer; 2012;69:3613–34.
  8. Tang YC, Amon A. Gene Copy-Number Alterations: A Cost-Benefit Analysis. Cell. Cell Press; 2013;152:394–405.
  9. Liu Y, Beyer A, Aebersold R. On the Dependency of Cellular Protein Levels on mRNA Abundance. Cell. Cell Press; 2016. p. 535–50.
  10. Schwanhüusser B, Busse D, Li N, Dittmar G, Schuchhardt J, Wolf J, et al. Global quantification of mammalian gene expression control. Nature. Nature Publishing Group; 2011;473:337–42.
  11. Buccitelli C, Selbach M. mRNAs, proteins and the emerging principles of gene expression control. Nat. Rev. Genet. Nature Publishing Group; 2020. p. 630–44.
  12. Maier T, Güell M, Serrano L. Correlation of mRNA and protein in complex biological samples. FEBS Lett. No longer published by Elsevier; 2009. p. 3966–73.
  13. Gedeon T, Bokes P. Delayed protein synthesis reduces the correlation between mRNA and protein fluctuations. Biophys J. Cell Press; 2012;103:377–85.
  14. Perl K, Ushakov K, Pozniak Y, Yizhar-Barnea O, Bhonker Y, Shivatzki S, et al. Reduced changes in protein compared to mRNA levels across non-proliferating tissues. BMC Genomics. BioMed Central Ltd.; 2017;18:1–14.
  15. Payne SH. The utility of protein and mRNA correlation. Trends Biochem. Sci. Elsevier Current Trends; 2015. p. 1–3.
  16. Casaro M, Souza VR, Oliveira FA, Ferreira CM. OVA-Induced Allergic Airway Inflammation Mouse Model. Methods Mol Biol [Internet]. Methods Mol Biol; 2019 [cited 2021 Aug 24]. p. 297–301. Available from: https://pubmed.ncbi.nlm.nih.gov/30535706/
  17. Sethi GS, Naura AS. Progressive increase in allergen concentration abrogates immune tolerance in ovalbumin-induced murine model of chronic asthma. Int Immunopharmacol. Elsevier; 2018;60:121–31.
  18. Shilovskiy IP, Sundukova MS, Babakhin, Gaisina AR, Maerle A V., Sergeev I V., et al. Experimental protocol for development of adjuvant-free murine chronic model of allergic asthma. J Immunol Methods. Elsevier; 2019;468:10–9.
  19. YH K, YJ C, EJ L, MK K, SH P, DY K, et al. Novel glutathione-containing dry-yeast extracts inhibit eosinophilia and mucus overproduction in a murine model of asthma. Nutr Res Pract [Internet]. Nutr Res Pract; 2017 [cited 2021 Aug 19];11:461–9. Available from: https://pubmed.ncbi.nlm.nih.gov/29209456/
  20. Wei Y, Luo QL, Sun J, Chen MX, Liu F, Dong JC. Bu-Shen-Yi-Qi formulae suppress chronic airway inflammation and regulate Th17/Treg imbalance in the murine ovalbumin asthma model. J Ethnopharmacol. Elsevier; 2015;164:368–77.
  21. O’Beirne SL, Salit J, Kaner RJ, Crystal RG, Strulovici-Barel Y. Up-regulation of ACE2, the SARS-CoV-2 receptor, in asthmatics on maintenance inhaled corticosteroids. Respir Res. BioMed Central; 2021;22.
  22. Weathington N, O’Brien ME, Radder J, Whisenant TC, Bleecker ER, Busse WW, et al. BAL cell gene expression in severe asthma reveals mechanisms of severe disease and influences of medications. Am J Respir Crit Care Med. American Thoracic Society; 2019;200:837–56.
  23. Singhania A, Rupani H, Jayasekera N, Lumb S, Hales P, Gozzard N, et al. Altered Epithelial Gene Expression in Peripheral Airways of Severe Asthma. PLoS One. Public Library of Science; 2017;12.
  24. Michalovich D, Rodriguez-Perez N, Smolinska S, Pirozynski M, Mayhew D, Uddin S, et al. Obesity and disease severity magnify disturbed microbiome-immune interactions in asthma patients. Nat Commun. Nature Publishing Group; 2019;10.
  25. Ahmed MM, Zaki A, Alhazmi A, Alsharif KF, Bagabir HA, Haque S, et al. Identification and Validation of Pathogenic Genes in Sepsis and Associated Diseases by Integrated Bioinformatics Approach. Genes (Basel). MDPI; 2022;13:209.
  26. Markov A V., Kel AE, Salomatina O V., Salakhutdinov NF, Zenkova MA, Logashenko EB. Deep insights into the response of human cervical carcinoma cells to a new cyano enone-bearing triterpenoid soloxolone methyl: A transcriptome analysis. Oncotarget. 2019;10:5267–97.
  27. Lee SY, Kim S, Han K, Woong Choi J, Byung Chae H, Yeon Choi D, et al. Microarray analysis of lipopolysaccharide-induced endotoxemia in the cochlea. Gene. Elsevier; 2022;823:146347.

Reviewer 3 Report

With interest, I read the manuscript biomedicines-1683746.

I have to admit that usually, I do not like studies based bioinformatics done on public databases and I typically suggest a rejection even in journal with much lower IF than in the case of Biomedicines. There are many reasons for this, some of which I will discuss here.

However, even those this study utilizes public databases as well, it is very different from typical public database-based in silico investigations. This particular study uses public databases in a smart way, in its first phase to select the candidate genes to be experimentally tested (wet work) in its second phase; in the third phase, some bioinformatics is used again to expand the results.

Thus, overall, I find this work interesting, with a potential to be published relatively high, even in Biomedicines.

Specific comments (no special order):

C1. Independently of the fact if it is discussed somewhere else in the manuscript or not, the Authors should widely address in the Discussion limitations of the study, especially those related to using public databases rather than in-house datasets. Specifically, lack of full knowledge in the subject selection, sample type, sample homogeneity (see also further), and quality of the samples and their processing should be mentioned.

C2. Regarding the experimental part, did the Authors consider detailed analysis of cell composition of BAL, specifically leukocytes? In the case of OVA-induced allergic airway inflammation, substantial increase in BAL eosinophils would be expected, which is usually considered as a proof that the model was successful.

C3. In continuation, this BAL eosinophilia would be observed in comparison to the control group (no OVA-sensitization and no OVA-challenge; or groups, if also the group with OVA-sensitization but no OVA-challenge was included). When was the group of “healthy” animals sacrificed? 24 hours after OVA challenge (so that it could serve as a model control)? Four weeks after OVA challenge?

C4. Please, do not write “OVA-induced asthma” but “OVA-induced allergic airway inflammation (mimicking human allergic asthma)”.

C5. Were immunoglobulins and cytokines measured, especially in BAL?

C6. There, roughly grouping, three types of public datasets used, as mentioned in lines 102-108, 109-114, and 114-120. The first two are fully murine and the last one is fully human, right?

C7. Regarding your own experiments and public mouse datasets (lines 102-108, 109-114), full lungs but not isolated/sorted cells were used, right? Please, make it clear regarding public datasets and describe as a limitation.

C8. By occasion, already in this part of the text (lines 102-114), please, make it very clear which datasets are human and which murine.

C9. Part of the bioinformatics analyses referring to lines 114-120, are presented first in the Discussion but not in the Results, right? I agree with this concept as it is only the expansion of your basic data using human data.

C10. Table 1 (with the primers) is methodological and should go to the supplement.

C11. Line 499. “Table 2” or “Table S2”?

C12. Names of the genes should be always written in italics, including tables and figures.

C13. Likewise, please, verify if human nomenclature is always used for human genes and murine for murine genes.

C14. The role of epithelial cells in COPD (PMID: 35326501) and asthma (PMID: 31904412) should be highlighted in the Introduction and/or the Discussion.

C15. Please, refer in the Discussion to the some recently published data (PMID: 33498209).

C16. Some coagulation-related mechanisms are overrepresented in your data. Please, mention that the established links between asthma and coagulation exist (PMID: 30053974, 24315352).

C17. Chemokine-related pathways are strongly represented in your data. Please, refer to PMID: 33918621 in your Discussion.

Author Response

Reviewer 3.

With interest, I read the manuscript biomedicines-1683746.

I have to admit that usually, I do not like studies based bioinformatics done on public databases and I typically suggest a rejection even in journal with much lower IF than in the case of Biomedicines. There are many reasons for this, some of which I will discuss here.

However, even those this study utilizes public databases as well, it is very different from typical public database-based in silico investigations. This particular study uses public databases in a smart way, in its first phase to select the candidate genes to be experimentally tested (wet work) in its second phase; in the third phase, some bioinformatics is used again to expand the results.

Thus, overall, I find this work interesting, with a potential to be published relatively high, even in Biomedicines.

Specific comments (no special order):

C1. Independently of the fact if it is discussed somewhere else in the manuscript or not, the Authors should widely address in the Discussion limitations of the study, especially those related to using public databases rather than in-house datasets. Specifically, lack of full knowledge in the subject selection, sample type, sample homogeneity (see also further), and quality of the samples and their processing should be mentioned.

Dear Reviewer #3,

Thank you very much for the high appreciation of our work and valuable comments that helped us to improve our manuscript. We revised and modified the manuscript according to your comments (revised parts are marked by red).

We agree that usage of public cDNA microarray datasets comes with a set of inherent limitations and challenges, specifically, the lack of knowledge about the subject/patient selection as well as type, homogeneity and quality of the samples. These moments, along with limitations regarding other parts of our work, were added in the limitations paragraph at the end of the Discussion section, please, see lines 871-896.

C2. Regarding the experimental part, did the Authors consider detailed analysis of cell composition of BAL, specifically leukocytes? In the case of OVA-induced allergic airway inflammation, substantial increase in BAL eosinophils would be expected, which is usually considered as a proof that the model was successful.

The development of OVA-induced asthma in mice leads to a significant increase in both eosinophils and neutrophils in the BAL fluid as well as mixed inflammatory infiltration in the lungs [1–4]. So we performed differential leukocyte counting of lymphocytes, granulocytes and monocytes subpopulations in order to prove the development of severe inflammation in the respiratory system of mice, which could further lead to pulmonary fibrosis development regardless of allergic or non-allergic etiology. It should also be noted, that unfortunately the staining method used in the study does not allow distinguishing reliably eosinophils, neutrophils and basophils in the granulocyte subpopulation.

C3. In continuation, this BAL eosinophilia would be observed in comparison to the control group (no OVA-sensitization and no OVA-challenge; or groups, if also the group with OVA-sensitization but no OVA-challenge was included). When was the group of “healthy” animals sacrificed? 24 hours after OVA challenge (so that it could serve as a model control)? Four weeks after OVA challenge?

Healthy group is represented by animals without Al(OH)3/OVA sensitization, without OVA challenge and used as a control. Healthy mice were sacrificed at the time point corresponding 24 h after the last OVA challenge of experimental mice (together with mice with acute asthma). We believe it is inappropriate to sacrifice additional group of healthy mice 4 weeks after the last OVA challenge (together with mice with pulmonary fibrosis), because it will be exactly the same mice as healthy group mentioned above kept under the same conditions (without both Al(OH)3/OVA sensitization and OVA challenge).

C4. Please, do not write “OVA-induced asthma” but “OVA-induced allergic airway inflammation (mimicking human allergic asthma)”.

We agree that the term “OVA-induced allergic airway inflammation” more closely describes the actual process in the lungs and airways of OVA-challenged mice, and we have replaced the term “OVA-induced asthma” with “OVA-induced allergic airway inflammation” where it is mentioned for the first time (please, see lines 70-71). However, in order to improve the manuscript readability, in all following cases the term “OVA-induced allergic airway inflammation” was referred to as “OVA-induced asthma” since we could not find a suitable short synonym of the mentioned term, and, moreover, “OVA-induced asthma” is a widespread term in scientific texts describing the type of murine asthma model very similar to the one used in this study [5–7].

C5. Were immunoglobulins and cytokines measured, especially in BAL?

Corrected. Unfortunately, we were unable to measure immunoglobulins due to technical limitations, but we have evaluated the levels of pro-inflammatory cytokines TNF-a and IL-6 in BAL fluid by ELISA; data were incorporated in the Materials and Methods section, Figure 3 and the main manuscript. Please, see Figure 3C and lines 158-161, 451-452, 466-467.

C6. There, roughly grouping, three types of public datasets used, as mentioned in lines 102-108, 109-114, and 114-120. The first two are fully murine and the last one is fully human, right?

Indeed, datasets concerning acute asthmatic inflammation in lungs and bleomycin-induced pulmonary fibrosis are fully murine, as well as datasets concerning chronic lung pathology are fully human. To avoid some confusion we included information about the species (“human” or “mouse”) of all datasets in the main manuscript (please, see lines 84-85, 91-92, 98) and in Table S1.

C7. Regarding your own experiments and public mouse datasets (lines 102-108, 109-114), full lungs but not isolated/sorted cells were used, right? Please, make it clear regarding public datasets and describe as a limitation.

You are indeed right. Full lung tissues but not their sorted cellular populations were used in both our experiments and public mouse transcriptomics analysis. We included information about the tissue/material in the description of all microarray datasets (data is presented in Table S1) and in the limitations paragraph at the end of the Discussion section (please, see Table S1 and lines 872-880).

C8. By occasion, already in this part of the text (lines 102-114), please, make it very clear which datasets are human and which murine.

Corrected. Thank you for your comments on unclear indications of species used for transcriptomics analysis. Due to the high number of analyzed datasets, indeed, it is easy to get confused about their characteristics. To fix this issue, we included information about the species (“human” or “murine”) of all datasets in the main manuscript (please, see lines 84-85, 91-92, 98) and in Table S1.

C9. Part of the bioinformatics analyses referring to lines 114-120, are presented first in the Discussion but not in the Results, right? I agree with this concept as it is only the expansion of your basic data using human data.

Yes, you are absolutely right. The exploration of the expression of human orthologs of revealed murine asthma/fibrosis-associated key genes in human chronic lung pathologies was first described in the Discussion section (lines 820-863). This analysis was performed only to understand whether their expression profiles are similar between different species and which human pulmonary disorder is more similar in this parameter to OVA-induced allergic airway inflammation in mice (translational meta-analysis). Thus, this paragraph is the expansion of our data described in the Results section.

C10. Table 1 (with the primers) is methodological and should go to the supplement.

Corrected. The Table containing information about primers used in the study was transferred to the Supplementary Materials (please, see Table S2).

C11. Line 499. “Table 2” or “Table S2”?

The data on the expression levels of key genes involved in both acute asthma and pulmonary fibrosis development (Fn1, Igf1, Ccl2, C3, Timp1, Cxcl12, Ccl4, Ccr2, Spp1, C3ar1, Cat, Cyp2e1, Muc5ac and Muc5b) revealed by bioinformatics analysis and evaluated in the lung tissue of OVA-challenged mice by qRT-PCR and immunohistochemistry (some of them) are summarized in Figure 4 and Table 2.

C12. Names of the genes should be always written in italics, including tables and figures.

Corrected. Thank you for this comment. This is our disappointing omission. We rewrote the names of the genes in italics in all figures where it was possible (please, see Figures 1, 2, 5, 6, and 8) except the following:

  • networks reconstructed by Cytoscape software (unfortunately, the settings of used plugins did not give opportunity to change the font styles within the networks; nevertheless, we emphasized in the figure captures that networks were reconstructed from “genes” or “DEGs” (please, see line 251 (Fig. 1C), line 337 (Fig. 2A), line 681 (Fig. 6A), line 819 (Fig. 7));
  • Discussion section (lines 772-813), since these parts of the manuscript concern both genes and their protein products of human and murine origin.

C13. Likewise, please, verify if human nomenclature is always used for human genes and murine for murine genes.

Corrected. We verified the human nomenclature for human genes and murine nomenclature for murine genes throughout the manuscript and Figures.

C14. The role of epithelial cells in COPD (PMID: 35326501) and asthma (PMID: 31904412) should be highlighted in the Introduction and/or the Discussion.

Corrected. We included the information about the role of epithelial cells in COPD and asthma development in the Introduction section. Please, see lines 36-38 and references [8] and [10].

C15. Please, refer in the Discussion to the some recently published data (PMID: 33498209).

We included the reference in the Introduction section. Please, see reference [6].

C16. Some coagulation-related mechanisms are overrepresented in your data. Please, mention that the established links between asthma and coagulation exist (PMID: 30053974, 24315352).

We included one of the suggested references in the Discussion section. Please, see reference [100].

C17. Chemokine-related pathways are strongly represented in your data. Please, refer to PMID: 33918621 in your Discussion.

The reference provided does not fully address the issue of the manuscript, so we decided not to include it.

Finally, to correct the text of our manuscript more thoroughly and to make the study more understandable, English editing was carried out using MDPI Language Editing Service and workflow of the study was added (Please, see Scheme 1).

References

  1. Wu HM, Fang L, Shen QY, Liu RY. SP600125 promotes resolution of allergic airway inflammation via TLR9 in an OVA-induced murine acute asthma model. Mol Immunol [Internet]. Mol Immunol; 2015 [cited 2022 Apr 13];67:311–6. Available from: https://pubmed.ncbi.nlm.nih.gov/26139014/
  2. Li X, Huang L, Wang N, Yi H, Wang H. Sulfur dioxide exposure enhances Th2 inflammatory responses via activating STAT6 pathway in asthmatic mice. Toxicol Lett [Internet]. Toxicol Lett; 2018 [cited 2022 Apr 13];285:43–50. Available from: https://pubmed.ncbi.nlm.nih.gov/29288730/
  3. Balkrishna A, Solleti SK, Singh H, Singh R, Bhattacharya K, Varshney A. Herbo-metallic ethnomedicine ‘Malla Sindoor’ ameliorates lung inflammation in murine model of allergic asthma by modulating cytokines status and oxidative stress. J Ethnopharmacol. Elsevier; 2022;292:115120.
  4. Bai D, Sun T, Lu F, Shen Y, Zhang Y, Zhang B, et al. Eupatilin Suppresses OVA-Induced Asthma by Inhibiting NF-κB and MAPK and Activating Nrf2 Signaling Pathways in Mice. Int J Mol Sci [Internet]. Int J Mol Sci; 2022 [cited 2022 Apr 13];23. Available from: https://pubmed.ncbi.nlm.nih.gov/35163503/
  5. Hanashiro J, Muraosa Y, Toyotome T, Hirose K, Watanabe A, Kamei K. Schizophyllum commune induces IL-17-mediated neutrophilic airway inflammation in OVA-induced asthma model mice. Sci Rep. Nature Publishing Group; 2019;9:1–9.
  6. Tian D, Yang L, Wang S, Zhu Y, Shi W, Zhang C, et al. Double negative T cells mediate Lag3-dependent antigen-specific protection in allergic asthma. Nat Commun. Nature Publishing Group; 2019;10:1–13.
  7. El-Hashim AZ, Khajah MA, Renno WM, Babyson RS, Uddin M, Benter IF, et al. Src-dependent EGFR transactivation regulates lung inflammation via downstream signaling involving ERK1/2, PI3Kδ/Akt and NFκB induction in a murine asthma model. Sci Rep. Nature Publishing Group; 2017;7:1–14.

Round 2

Reviewer 2 Report

The manuscript has some unusual formatting; please consider revision/reorganization.

  • There is a "Scheme 1", but also a separate "Figure 1"
  • There is a "Table 2", but no "Table 1"
  • Ln808, capitalize "Collagen ..."
  • References contain website links. These are overly long (e.g. Ref11) and there is no guarantee they still work after a couple years
  • Formatting of author's names in Ref47 is wrong

Author Response

Dear Reviewer #2,

We are sincerely grateful to you for your thorough analysis of the revised manuscript. The manuscript was corrected according to your comments (revised parts are marked by yellow).

Comments and suggestions for authors:

  • There is a "Scheme 1", but also a separate "Figure 1"

Corrected. We renamed “Scheme 1” to “Figure 1” and modified the numeration of other figures in the figure legends and throughout the manuscript.

  • There is a "Table 2", but no "Table 1"

Corrected. Sorry, it is our regrettable misprint. We renamed “Table 2” to “Table 1” in the table caption and throughout the manuscript.

  • Ln808, capitalize "Collagen ..."

Corrected. Please, see line 808.

  • References contain website links. These are overly long (e.g. Ref11) and there is no guarantee they still work after a couple years

Corrected. We reviewed and modified all references according Biomedicines reference style.

  • Formatting of author's names in Ref47 is wrong

Corrected. We modified the authors’ name in Ref47 in a right format.

Reviewer 3 Report

My comments have been addressed well. Thank you.

Author Response

Dear Reviewer #3,

Thank you very much for your deep analysis of our manuscript and highly valuable comments. We hope that our findings will be useful for other research groups investigating pulmonary inflammatory disorders and fibrotic remodeling.
